# Insights into the genome of *Methylobacterium* sp. NMS14P, a novel bacterium for growth promotion of maize, chili, and sugarcane

Jiraporn Jirakkakul[1☯], Ahmad Nuruddin Khoiri[2☯], Thanawat Duangfoo[1], Sudarat Dulsawat[1], Sawannee Sutheeworapong[1], Kantiya Petsong[1,3], Songsak Wattanachaisaereekul[1,4], Prasobsook Paenkaew[1], Anuwat Tachaleat[1], Supapon Cheevadhanarak[1,2]*, Peerada Prommeenate[5]*

1 Pilot Plant Development and Training Institute, King Mongkut's University of Technology Thonburi, Bangkok, Thailand, 2 Bioinformatics and Systems Biology Program, School of Bioresources and Technology, King Mongkut's University of Technology Thonburi, Bangkok, Thailand, 3 Department of Food Technology, Faculty of Technology, Khon Kaen University, Khon Kaen, Thailand, 4 School of Food Industry, King Mongkut's Institute of Technology Ladkrabang, Bangkok, Thailand, 5 Biochemical Engineering and Systems Biology Research Group, National Center for Genetic Engineering and Biotechnology, National Science and Technology Development Agency at King Mongkut's University of Technology Thonburi, Bangkok, Thailand

☯ These authors contributed equally to this work.
* peerada.pro@biotec.or.th (PP); supapon.che@kmutt.ac.th (SC)

**Data Availability Statement:** The assembled genome and raw reads have been submitted to NCBI under Bioproject number PRJNA780213

## Abstract

A novel methylotrophic bacterium designated as NMS14P was isolated from the root of an organic coffee plant (*Coffea arabica*) in Thailand. The 16S rRNA sequence analysis revealed that this new isolate belongs to the genus *Methylobacterium*, and its novelty was clarified by genomic and comparative genomic analyses, in which NMS14P exhibited low levels of relatedness with other *Methylobacterium*-type strains. NMS14P genome consists of a 6,268,579 bp chromosome, accompanied by a 542,519 bp megaplasmid and a 66,590 bp plasmid, namely pNMS14P1 and pNMS14P2, respectively. Several genes conferring plant growth promotion are aggregated on both chromosome and plasmids, including phosphate solubilization, indole-3-acetic acid (IAA) biosynthesis, cytokinins (CKs) production, 1-aminocyclopropane-1-carboxylate (ACC) deaminase activity, sulfur-oxidizing activity, trehalose synthesis, and urea metabolism. Furthermore, pangenome analysis showed that NMS14P possessed the highest number of strain-specific genes accounting for 1408 genes, particularly those that are essential for colonization and survival in a wide array of host environments, such as ABC transporter, chemotaxis, quorum sensing, biofilm formation, and biosynthesis of secondary metabolites. *In vivo* tests have supported that NMS14P significantly promoted the growth and development of maize, chili, and sugarcane. Collectively, NMS14P is proposed as a novel plant growth-promoting *Methylobacterium* that could potentially be applied to a broad range of host plants as *Methylobacterium*-based biofertilizers to reduce and ultimately substitute the use of synthetic agrochemicals for sustainable agriculture.

(BioSample accession number: SAMN23012515; SRA accession number for the Nanopore read: SRR19445089; SRA accession number for Illumina reads: SRR19445090).

**Funding:** AK was financially supported by the "Scholarship for the Development of High Quality Research Graduates in Science and Technology Petchra Pra Jom Klao Ph.D. Research Scholarship (KMUTT – NSTDA) from King Mongkut's University of Technology Thonburi". KP was funded by The Program Management Unit for Human Resources & Institutional Development, Research and Innovation [grant number B01F630003]. This research project is supported by Thailand Science Research and Innovation (TSRI) Basic Research Fund: Fiscal year 2022 under project number FRB650048/0164. The funders had no role in study design, data collection and analysis, decision to publish, or preparation of the manuscript.

**Competing interests:** The authors have declared that no competing interests exist.

## Introduction

The use of agrochemicals has enormous adverse effects on the environment, human and animal health, and soil biodiversity, threatening food security and safety globally. The continuous accumulation of agrochemical residues in soil reverses soil microbial diversity, resulting in the loss of soil fertility, which cannot sustain plant growth and development [1, 2]. To diminish agrochemical usage, plant growth-promoting bacteria (PGPB) are proposed as an alternative agent or "biofertilizer" to promote the resilience of the agroecosystem. The term "biofertilizer" has been applied for the use of living microorganisms which, when applied to seeds, plant surfaces, or soils, can live together with plants and stimulate the growth of the host by increasing the bioavailability of primary nutrients and providing beneficial traits [3]. Important characteristics of several PGPB include their capability to perform biological nitrogen fixation, phosphate solubilization, 1-aminocyclopropane-1-carboxylic acid (ACC) deaminase activity, and production of siderophores and phytohormones [4]. The interest in applying PGPB as biofertilizers has been increasing as a sustainable substitute for the use of agrochemicals in crop production. However, ecological and geographical factors have been known to have a significant impact on the sustainable establishment of PGPB [5]. Therefore, the discovery of the local strains could be a key success factor for substantial use as biofertilizers that are eco-friendly and improve soil fertility for each specific region.

Several members of the genus *Methylobacterium* have been reported for their plant growth-promoting (PGP) activities, including *M. oryzae*, *M. nodulans*, *M. radiotolerans*, and some unclassified species of the *Methylobacterium* genus [6–8], implying their importance as microbial-based biofertilizers. They can utilize organic one-carbon (C1) compounds as the sole source of carbon and energy, especially methanol. In association with plants, *Methylobacterium* gains benefit from methanol which is emitted by the host plant as a by-product of the cell wall synthesis [9]. *Methylobacterium* is normally found in the phyllosphere, however, some have been isolated from roots, shoots, seeds, and seedlings [10]. In nature, *Methylobacterium* usually co-occurs with other bacteria such as *Pseudomonas* spp. and *Sphigomonas* spp. [11] and arbuscular mycorrhizal fungi (AMF) [12]. This indicates that *Methylobacterium* commonly exists as a member of a consortium that can adapt to a variety of environmental conditions and perform several tasks of plant growth stimulation, i.e., plant nutrient support, plant hormone production, and pathogen inhibition [13, 14].

In this study, *Methylobacterium* spp. were isolated from organic coffee roots by imprinting methods. One of the isolates designated as *Methylobacterium* sp. NMS14P (hereafter referred to as NMS14P) revealed significant PGP activities as observed from the various plant phenotypes (maize, chili, and sugarcane), which included root development, height, and total biomass. Maize, chili, and sugarcane are economical crops of Thailand, which require a lot of chemical fertilizers and pesticides to increase their productivity. Therefore, the success in the application of NMS14P as a potential native PGPB could provide an alternative way for sustainable agriculture management. To evaluate the functional capabilities of NMS14P as a PGPB, the whole genome of NMS14P was sequenced and comparatively analyzed with 11 closely related *Methylobacterium* genomes. The overall genomic information of NMS14P supports and provides insight into the molecular mechanisms underlying the plant growth-promoting characteristics of NMS14P, which could be applied to the development of either a single or consortium of *Methylobacterium*-based biofertilizers in the future.

## Materials and methods

### Isolation of NMS14P

NMS14P was isolated from the root of coffee (*Coffea arabica* variety Chiang Mai 80) during the reproductive phase, employing the root-imprinting method as described by Corpe (1985) [15] with some modifications as follows. The coffee roots were collected from an organic coffee plantation at Baan Pa Miang, Lampang, Thailand. The soil particles were removed from the roots manually, and the roots were then washed with sterile distilled water 5 times. Subsequently, they were air-dried before being placed on nitrate mineral salts agar medium (NMS; ATCC medium 1306) supplemented with 1% (v/v) methanol and cycloheximide (100 μg/ml) to inhibit the growth of fungi. After incubating the plates at 30˚C for 7 days, the pink-pigmented bacterial colonies were picked and purified by re-streaking on NMS agar supplemented with 1% (v/v) methanol until a single isolate was obtained. The selected colony designated as NMS14P was then kept in 20% glycerol for further characterization.

### Preparation of NMS14P inoculum

NMS14P was grown on NMS agar supplemented with 1% methanol at 30˚C. After 7 days of incubation, the NMS14P lawn was scraped and resuspended in sterile distilled water. The cell suspension was inoculated into a 1/4 strength of nutrient broth (Difco, USA) at OD600 of 0.2 and cultured to the mid-log phase at 25˚C with shaking at 150 rpm. The cell pellet was collected by centrifugation at 8,000g for 15 min, washed twice, and resuspended in sterile distilled water. NMS14P solution at OD600 of 0.3 (approximately $10^8$ CFU/ml) was used as an inoculum at a final concentration of $10^6$ CFU per gram of sterile soil.

### Effects of NMS14P inoculation on the growth of maize (*Zea mays* L.), chili (*Capsicum annuum*), and sugarcane (*Saccharum officinarum* L.)

The effect of NMS14P inoculation on plant growth was observed by using maize, chili, and sugarcane as plant models. The pot experiments were conducted in a controlled growth room (26±4˚C temperature, 72±5% relative humidity, 176.56±47.09 $\mu mol.m^{-2}.s^{-1}$ brightness, and 16/8-h light/dark photoperiod), located at the King Mongkut's University of Technology Thonburi, Bangkhuntien Campus. All plants were grown in a soil mixture consisting of peat moss and perlite at a 1:1 (v/v) ratio and then inoculated with NMS14P suspension, defined as a treatment, while the controls were treated with sterile distilled water. For maize, the seeds (genotype PAC339; Pacific Seeds, Thailand) were surface sterilized by rinsing with 150 ml of sterile distilled water (1 min, 3 times), and soaking for 5 min in 100 ml of 2% sodium hypochlorite (NaClO) solution (Sigma Aldrich, USA), then rinsing with 150 ml of sterile distilled water, followed by immersing in 70% (v/v) ethanol (Merck Company Ltd., Germany) for 5 min. The final wash was performed by rinsing with 150 ml of sterile distilled water for 5 min, 5 times. The seeds were air-dried in a biosafety cabinet and subsequently germinated in 6 x 5-inch pots (height x diameter) for approximately 7 days before the treatment. For chili, the chili seeds (variety Superhot 2 F1; East-West Seed, Thailand) were surface sterilized by immersing in 70% (v/v) ethanol for 1 min and followed by immersing in 2% NaClO solution (Sigma Aldrich, USA) for 20 min. Seeds were finally washed in sterile distilled water 5 times before air-dried in a biosafety cabinet. The sterilized seeds were sown in a 30 g sterile peat moss: perlite mixture (1:1; v/v) in a 50 ml falcon tube. After 4 weeks of germination, seedlings were transferred to 4 x 6-inch pots (height x diameter) filled with 350 g of sterile peat moss: perlite mixture (1:1; v/v). For sugarcane, the plantlets (variety Khon Kaen 3 (KK3); obtained from the plant propagation center No.10 Udon Thani Province, Thailand) were transferred to

peat moss: perlite mixture (1:1; v/v) in 6 x 5-inch pots (height x diameter) and grown for approximately 4 weeks before the treatment. The *in vivo* experiments were carried out with 3, 4, and 5 replicates for sugarcane, chili, and maize, respectively.

The plants were regularly watered every two days with 100 ml of sterile distilled water. NMS14P suspension at a final concentration of $10^6$ CFU/g soil was applied as a soil inoculum during the first week of plantation and every two weeks until the end of the experiments. The sterile distilled water was used in the control plants. Plant growth was determined by measuring cumulative plant height, number of leaves, and shoot and root dry weight at 35, 75, and 56 days after inoculation for maize, chili, and sugarcane, respectively. Shoot and root dry weights were obtained after oven drying at 70˚C until a constant weight was achieved.

Statistical analysis of plant growth-promoting experiment was carried out by t-test using R program v.3.6.3 [16] and subsequently visualized with *ggplot2* R package [17].

## Inorganic phosphate solubilization test on Pikovskaya's (PVK) agar

NMS14P was streaked on Pikovskaya's (PVK) agar plate containing tricalcium phosphate as an inorganic phosphate source and incubated at 30˚C for 7 days. Inorganic phosphate solubilization activity was observed from the clear zone around the colony [18].

## Alkaline phosphatase activity assay

The alkaline phosphatase activity of NMS14P was determined using the Alkaline Phosphatase Yellow (pNPP) Liquid Substrate System for ELISA (Sigma-Aldrich) following the instruction from the manufacturer with some modifications as follows. A total of 200 ml of NMS14P suspension was centrifuged and the supernatant was discarded. The cell pellet was resuspended in 400 μl of pNPP substrate and incubated for 24 hours in a dark condition at room temperature. The reaction was stopped with 100 μl of 3M NaOH solution. The reaction mix was centrifuged to discard the bacterial cells, and supernatant color was observed. The presence of alkaline phosphatase activity was determined from the yellow color of the reaction compared to the colorless of the control, which did not contain bacterial cells [19].

## Urease activity assay

NMS14P was grown on NMS agar supplemented with 1% methanol at 30˚C for 7 days. The lawn of the colony was scraped and suspended in sterile distilled water. Urease activity was determined by the colorimetric method following Tanaka et al. (2003) [20] with some modifications. A urea buffer solution containing 3% urea in phosphate buffer with pH 7 and 0.001% phenol red as an indicator was prepared. NMS14P cell was then suspended in this urease buffer solution and incubated at room temperature for 24 hours. The color change was observed by comparing it with the control reaction which had no bacterial cells. The color change from light orange to magenta in the urea buffer solution indicated positive urease activity.

## 1-aminocyclopropane-1-carboxylic acid (ACC) deaminase activity test

ACC deaminase activity was determined following Penrose and Glick (2008) [21] with some modifications. NMS14P was cultured in a Dworkin-Foster (DF) salt minimal medium supplemented with 5 mM $(NH_4)_2SO_4$ as a nitrogen source for non-ACC-induced control and 3.0 mM ACC as a nitrogen source for ACC-induced cells. The ACC solution was sterilized through a 0.2 μM membrane filter. Cell suspension of NMS14P was inoculated into the medium and incubated at 25˚C with shaking at 150 rpm for 7 days. A 5 ml cell was harvested

by centrifugation at 8,000 g for 10 min. The cells were then washed with 0.1 M Tris-HCl, pH 7.6. A 1 ml cell suspension was transferred to a 1.5 ml microcentrifuge tube and centrifuged at 13,000 g for 5 min before resuspending in 600 μl of 0.1 M Tris-HCl, pH 8.5. The cell suspension was extracted by adding 30 μl of toluene and vortexed for 30 s. The toluenized cell suspension was subjected to an ACC deaminase activity test. To detect the by-product, α-ketobutyrate, of the ACC deaminase reaction, 20 μl of 0.5 M ACC was added to the 200 μl of toluenized cell suspension, and then incubated at 30°C for 15 min. After that, 1 ml of 0.56 N HCl was added to the reaction, vortexed, and then centrifuged for 5 min at 13,000 rpm at room temperature. A 1 ml supernatant was mixed with 800 μl of 0.56 N HCl in a glass tube and added with 300 μl of 2,4-dinitrophenylhydrazine reagent (0.2% 2,4-dinitrophenylhydrazine in 2 N HCl), then vortexed before incubated at 30°C for 30 min. A total of 2 ml of 2 N NaOH was added and mixed. The reaction containing α-ketobutyrate and 2,4-DNP-hydrazone produced a brown solution. The amount of α-ketobutyrate from the reaction was determined by measuring its absorbance at 540 nm and compared with the blank control and non-ACC-induced cell [21].

## Indole acetic acid (IAA) biosynthesis determination

To determine the IAA biosynthesis of NMS14P, a single colony of NMS14P was inoculated in 5 ml of half-strength Tryptic Soy broth (TSB) supplemented with 5, 10, and 20 mM L-tryptophan and incubated at 25°C with shaking at 150 rpm for 6 days. The supernatant was collected by centrifugation at 8,000 g for 10 min. IAA was detected by the colorimetric method as described by Gilbert et al. (2018) [22] with slight modifications. The Salkowski reagent was added to the supernatant at a 2:1 ratio (v/v) (Salkowski reagent: Supernatant) in triplicate. The reactions were incubated in a dark condition at room temperature for 30 min. The concentration of IAA production was measured at the wavelength of 530 nm and then compared with the IAA standard curve.

## Hypersensitivity reaction (HR) assay

Hypersensitivity reaction (HR) assay was performed as described by Huang et al. (1988) [23] with some modifications. Briefly, the bacterial cell suspension at the OD600 of 0.3 was prepared in sterile distilled water. Tomato (*Solanum lycopersicum* L.) leaves were wounded by a 27-gauge needle underneath the leaves and then infused with 10 μl of sterile distilled water, *Pseudomonas aeruginosa* suspension, and *Methylobacterium* sp. MNS14P suspension for the negative control, positive control, and NMS14P, respectively. The infiltrated leaves were put on wet cotton in a sterilized 90 mm petri dish and incubated at room temperature (12-h light/dark cycle). The development of HR was observed every day.

## Genomic DNA extraction and sequencing

Genomic DNA (gDNA) of NMS14P was extracted using the ZymoBIOMICS MagBead DNA kit according to the manufacturer's instructions (Zymo Research, USA). The quality and quantity of the extracted gDNA were further examined with a NanoDrop spectrophotometer (Thermo Scientific, Wilmington, DE, United States). A total of 2 μg of genomic DNA was sent to Novogen AIT Singapore for paired-end (2 x 150 bp) sequencing using a Nextera XT sequencing kit on Illumina NovaSeq 6000. For nanopore long-read sequencing, 1.2 μg of genomic DNA was subjected to library preparation using an SQK-LSK109 ligation kit (Nanopore, Oxford, UK), and sequencing was performed in-house on the MinION R9 flow cell (Nanopore, Oxford, UK), according to the manufacturer's protocol.

## Quality control for sequencing reads

Low-quality bases and adapter sequences of Illumina reads were filtered out using fastp v.0.20.1 [24] with parameters "–qualified_quality_phred 25 and–unqualified_percent_limit 20". For Nanopore, MinION raw reads in the format of "FAST5" were converted into "FASTQ" files using Albacore v.2.3.4 (https://community.nanoporetech.com). The converted FASTQ reads with a length shorter than 500 bases and an average quality score below 7 were discarded using NanoFilt v.2.7.1 [25]. Sequencing adapters were further filtered using Porechop v.0.2.3 (https://github.com/rrwick/porechop).

## Genome assembly and quality assessment

The genome assembly was done using Trycycler v.0.4.1 [26]. In brief, filtered MinION reads were subsampled into 12 different read sets and further assembled with Flye v.2.8.2-b1689 [27], Miniasm v.0.3-r179 [28], Minipolish v.0.1.2 [29], and Raven v.1.3.0 [30]. The contigs were then clustered into per-replicon groups to remove spurious, incomplete, or misassembled contigs and reconciled to fix circularization issues. Subsequently, multiple sequence alignment and partitioning reads were performed to generate consensus contigs. Finally, the assembled genome was polished with Illumina paired-end reads using Pilon v.1.23 [31]. The quality of assembly was assessed using CheckM v.1.0.12 [32], QUAST v.5.1.0rc1 [33], and BUSCO v.4.1.4 [34].

## Genome annotation and visualization

Genome annotation was carried out using Prokka v.1.14.5 [35]. Briefly, protein-coding sequences (CDSs) were predicted using prodigal v.2.6.3 [36]. rRNA, tRNA, and ncRNA were identified with Barrnap v.0.9 (https://github.com/tseemann/barrnap), ARAGORN v.1.2.38 [37], and Infernal v.1.1.2 [38], respectively. Functional annotation of every CDS was then performed with the following tools and databases. BLAST+ [39] implemented in Prokka was used to conduct sequence similarity searching with a custom database comprising all protein sequences belonging to the genus *Methylobacterium* retrieved from the NCBI database as of August 2021. Kyoto Encyclopedia of Genes and Genomes (KEGG) annotation was done using BlastKOALA v.2.2 [40]. Clusters of Orthologous Groups (COG) profiles were determined using DIAMOND BLASTp v.0.9.24 [41] against the COG database [42], downloaded from NCBI as of August 2021. DIAMOND BLASTp v.0.9.24 [41] was also employed for aligning predicted CDSs against NCBI-nr (downloaded as of August 2021). In addition, RAST server v.2.0 [43] was also utilized for automatic bacterial genome annotation. The carbohydrate-active enzyme (CAZy) prediction was executed using the dbCAN2 meta server [44]. Circular genome and plasmid visualizations were performed with CGView v.1.0 [45].

## Genomic islands, pathogenic potential, virulence, and antimicrobial resistance (AMR) genes predictions

Genomics islands were computationally predicted using IslandViewer 4 [46]. The pathogenic potential of NMS14P was evaluated using PathogenFinder v.1.1 web service with automated mode [47]. The putative virulence factors were assessed using BLASTp search against protein sequences from the full dataset in the virulence factor database (VFDB) [48]. The presence of AMR genes was identified with a web server of Resistance Gene Identifier (RGI) v.5.1.1 module of Comprehensive Antibiotic Resistance Database (CARD) v.3.1.1 [49].

## Bacterial species identification and classification

Bacterial species identification was performed using 16S rRNA gene and overall genome relative index (OGRI) methods. For 16S rRNA-based identification, it was done by directly amplifying the 16S rRNA gene from the NMS14P single isolated colony by colony PCR using universal primers 27F and 1492R (27F: 5'-AGAGTTTGATCCTGGCTCAG-3' and 1492R: 5'-GGTTACCTTGTTACGACTT-3'). PCR product was further purified and sequenced with the Sanger sequencing technique with the 1492R primer. The output sequence was inspected, and low-quality bases were trimmed using Sequence Scanner Software v2.0 (ThermoFisher Scientific) prior to downstream analyses (S1 Text). A high-quality sequence was then used as a query for blast search using NCBI BLASTn (https://blast.ncbi.nlm.nih.gov). Additionally, the Sanger sequencing-derived 16S rRNA gene sequence was also used to ensure the authenticity of the genome assembly by pairwise alignment with the predicted 16S rRNA sequences from the assembled genome [50]. Taxonomic classification based on OGRI was done with two approaches as follows. Firstly, a whole-genome-based taxonomic analysis was carried out using the Type (Strain) Genome Server (TYGS), an open-source bioinformatics platform available under https://tygs.dsmz.de [51]. 16S rRNA-based phylogenetic and phylogenomic trees were inferred with BLAST distance as implemented in TYGS and visualized using the *ggtree* R package [52]. Secondly, a core genome alignment method as proposed by Chung et al. (2018) [53] was employed to identify a set of orthologous sequences conserved in all aligned genomes. In addition, the genomic similarities between strain NMS14P and other closely related species (Table 1) were determined using the average nucleotide identity (ANI) algorithm with FastANI v.1.32 [54]. Core genome alignment similarity index (CGASI) and ANI matrices were visualized with the *ggplot2* R package [17].

## Comparative genomic analysis

To avoid comparison biases caused by different annotation tools, all reference genomes downloaded from Refseq NCBI (Table 1) were re-annotated with Prokka v.1.14.5 [35]. Pangenome analysis was subsequently carried out using Roary v.3.13.0 [55] with the minimum percentage identity for BLASTp set to 90. Single representative sequences from each of the clusters in the pangenome were further aligned against the KEGG database using GhostKOALA [40], while COG and custom *Methylobacterium* database annotations were executed with DIAMOND BLASTx v.0.9.24 [41]. The analysis results were then visualized using the R program v.3.6.3 [16]. The comparison of 12 *Methylobacterium* genomes was executed with CGView Comparison Tools (CCT) v.1.0.2 [56].

The overall method performed in this study was summarized in a flowchart and is provided in the S1 Fig.

## Results

### *In vivo* plant growth-promoting activity test of NMS14P

NMS14P was isolated from the organic coffee roots by the root imprinting method. To assess the PGP traits of NMS14P, *in vivo* tests were carried out with three economically important plants, namely maize, chili, and sugarcane under a greenhouse environment (S2 Fig). As shown in Fig 1A, the shoot dry weight and shoot height of NMS14P-inoculated maize were significantly greater than the control plants ($p < 0.05$). In chili, inoculation of NMS14P revealed significantly higher shoot dry weight and the number of leaves than the control plants ($p < 0.05$) (Fig 1B). NMS14P also significantly stimulated the growth of sugarcane in terms of cumulative plant height when compared with control plants ($p < 0.05$) (Fig 1C). These results

**Table 1. Reference genomes used for comparative genomic analysis.**

| GenBank assembly accession | Taxonomy | Assembly level | Number of contigs | Genome size (bp) | GC content (%) | CDS |
|---|---|---|---|---|---|---|
| GCA_001936175 | *Methylobacterium phyllosphaerae* CBMB27 | Complete | 4 | 6,316,624 | 69,57 | 5,774 |
| GCA_003254375 | *Methylobacterium* sp. XJLW | Complete | 2 | 6,666,616 | 69,89 | 6,246 |
| GCA_000757795 | *Methylobacterium oryzae* CBMB20 | Complete | 5 | 6,524,597 | 69.53 | 6,029 |
| GCA_001854385 | *Methylobacterium* sp. C1 | Complete | 1 | 6,459,145 | 71,23 | 6,077 |
| GCA_000019725 | *Methylobacterium radiotolerans* JCM 2831 | Complete | 9 | 6,899,110 | 71,04 | 6,495 |
| GCA_003096615 | *Methylobacterium organophilum* DSM 760 | Scaffold | 85 | 6,750,984 | 71,38 | 6,280 |
| GCA_014138435 | *Methylobacterium fujisawaense* DSM 5686 | Scaffold | 25 | 5,971,667 | 70,03 | 5,545 |
| GCA_007991055 | *Methylobacterium radiotolerans* NBRC 15690 | Scaffold | 170 | 6,791,170 | 71,1 | 6,416 |
| GCA_004011495 | *Methylobacterium radiotolerans* ES_PA-B5 | Contig | 9 | 7,696,971 | 70,67 | 7,205 |
| GCA_903971015 | *Methylobacterium radiotolerans* ME94 | Contig | 166 | 6,414,050 | 71,47 | 5,954 |
| GCA_001981325 | *Methylobacterium radiotolerans* RE1.2 | Contig | 226 | 6,298,400 | 71,29 | 5,882 |

indicated that a single strain treatment of NMS14P could substantially promote the growth of both monocot and dicot plants of commercial importance. It was thus hypothesized that NMS14P might be able to provide extra available nutrients by mobilizing the essential building blocks from the soil and environments and function as a biofertilizer for plant growth and development. In addition, applications of NMS14P in maize, chili, and sugarcane under a greenhouse environment revealed the possible broad host range of PGP activities of this bacterial species. Besides PGP activities, the hypersensitivity reaction (HR) assay revealed that no necrotic response was found on the NMS14P-infiltrated leaves of the tomato plant model as shown in the S3 Fig. Altogether, it is interesting to investigate the molecular mechanisms underlying these PGP traits at the genomic level. Thus, the genome of NMS14P was further characterized to identify the functional capabilities of the genes governing its PGP attributes.

## General information on the genomic dataset

A total of 572,350 and 10,270,449 raw reads were generated by the Nanopore and Illumina sequencing platforms, respectively (S1 Table). After removing low-quality bases, 0.97% of Nanopore reads were filtered out, while 0.94% of reads were discarded for Illumina. The remaining sequences were 555,629 with an average Phred score of 9.3 and mean read length of 5,394.2 and 9,671,658 paired-end reads with an average Phred score of 36 and mean read length of 149 for Nanopore and Illumina, respectively (S1 Table). These high-quality sequences were used for further genome assembly steps.

## Genome assembly and quality assessment of NMS14P

The whole-genome sequences of NMS14P were obtained by the combination of Illumina paired-end and Nanopore sequencing platforms. Genome assembly was performed with Trycycler [26] by combining several assembler programs, including Flye [27], Miniasm [28], Minipolish [29], and Raven [30] to get a long-read assembly consensus, which was further polished with high-quality Illumina short-reads using Pilon [31]. Genome completeness and accuracy were estimated using BUSCO [34], CheckM [32], and QUAST [33]. The quality of the assembled genome revealed completeness of more than 99.5%, contamination of 0.63%, and 743X average depth coverage (S2 Table), implying a high-quality assembled genome. Moreover, a high identity of 16S rRNA sequences derived from Sanger sequencing compared with the ones extracted from the whole genome assembly (S1 Text) confirmed the authenticity of genomic data [50].

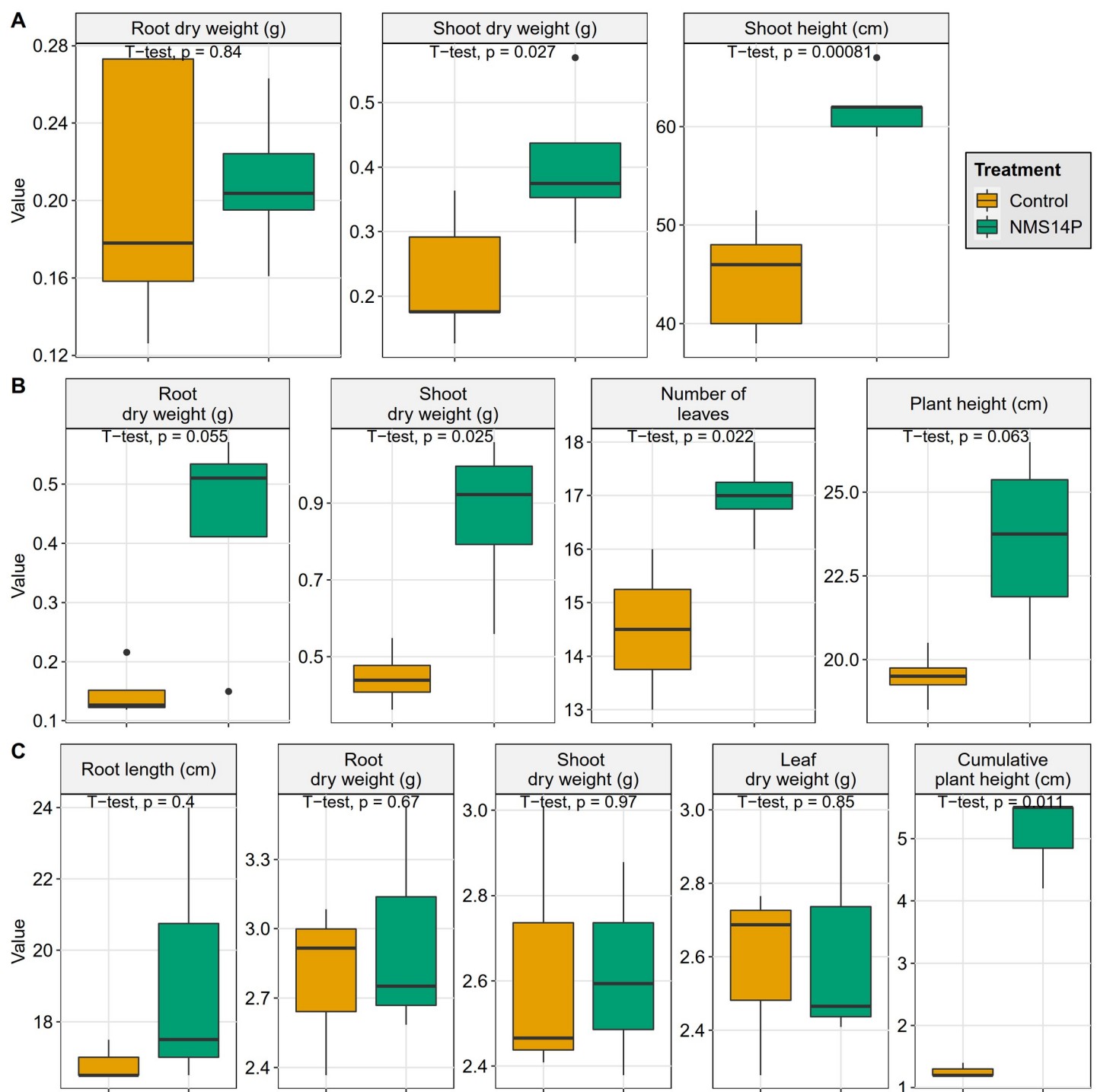

**Fig 1. *In vivo* plant growth-promoting activity test of NMS14P.** Greenhouse experiments to evaluate the potential of NMS14P as a PGPB were performed on (A) maize, (B) chili, and (C) sugarcane.

## General characteristics of the NMS14P genome

A high-quality complete genome was generated for NMS14P, comprising a single circular chromosome of 6,268,579 bp with a high G+C content of 71.12%, and two plasmids labeled as pNMS14P1 (542,519 bp; considered as a megaplasmid) and pNMS14P2 (66,590 bp) (Fig 2), possessing overall G+C contents of 70.61% and 64.62%, respectively (Table 2). A total of 5,879 protein-coding sequences (CDSs) with an average length of 914 bp, 12 rRNAs, and 73 tRNAs were predicted from the NMS14P chromosome, of which 918 (15.61%) of CDSs were assigned as proteins with unknown or hypothetical function (Table 2). Furthermore, 496 CDSs with an average length of 904 bp were identified in pNMS14P1, and 113 CDSs with an average length of 463 bp were found in pNMS14P2 (Table 2). Genome annotation results for NMS14P are provided in the S3 Table. The assembled genome and raw reads have been submitted to NCBI under Bioproject number PRJNA780213 (BioSample accession number: SAMN23012515; SRA accession number for the Nanopore read: SRR19445089; SRA accession number for Illumina reads: SRR19445090).

## Identification of NMS14P as a novel species

NMS14P was taxonomically identified using both full-length 16S rRNA gene and complete genome information, including digital DNA-DNA hybridization (dDDH), average nucleotide identity (ANI), and core genome alignment similarity index (CGASI). The position of NMS14P in both 16S rRNA-based phylogenetic and phylogenomic trees was consistent and revealed that this species was evolutionarily separated from the closely related *M. organophilum* DSM 760 and *M. radiotolerans* JCM 2831 (Fig 3). Furthermore, NMS14P showed low similarity scores with other *Methylobacterium*-type strains, which were less than the species cutoff of 70% and 95% for dDDH (formula d4) (Table 3) and ANI (Fig 4), respectively. NMS14P also exhibited CGASI values lower than the 96.8% identity threshold of bacterial species boundary (Fig 4) against 11 reference genomes in the same genus, and a paraphyletic clade was not observed (S4 Fig). Therefore, these results indicated that NMS14P is a new *Methylobacterium* species.

## Genomic islands and preliminary safety assessment of NMS14P

A total of 118 genomic islands (GIs) were identified by IslandViewer 4 [46] throughout the NMS14P genome (S4 Table). Particularly, 115 and 3 GIs were discovered in the chromosome (1,253,005 bp, 19.99% of the entire chromosome) and pNMS14P1 (19,182 bp, 3.54% of the overall pNMS14P1), respectively. The size of these putative islands ranged from 4,025 bp to 54,755 bp. The largest GI (GI-013) possessed 50 genes, whereas the smallest GI (GI-048) contained 12 genes. The majority of the predicted GIs were genes encoding for hypothetical proteins (811 genes), followed by tyrosine recombinase XerC (18 genes), sensor histidine kinase RcsC (12 genes), IS5 family transposase ISMex40 (10 genes), IS630 family transposase ISMex30 (9 genes), D-inositol-3-phosphate glycosyltransferase (7 genes), IS3 family transposase ISMtsp5 (7 genes), protein-methionine-sulfoxide reductase catalytic subunit MsrP (5 genes), and other products with numbers of genes less than 5 (S4 Table). Notably, three GIs, i.e., GI-059, GI-070, and GI-079, were recognized as genes to potentially confer the adaptability and competitive traits of this isolate in the rhizosphere environment.

The pathogenic potential of NMS14P was assessed using PathogenFinder [47]. This web-based tool predicted NMS14 as a non-human pathogen with a low probability score of 0.19 (S5 Table). The presence of virulence factors in the NMS14P genome was identified using the virulence factors of the pathogenic bacteria database (VFDB) [48]. Accordingly, 11 and 2 hits were found for genes designated as virulence factors located in the chromosome and

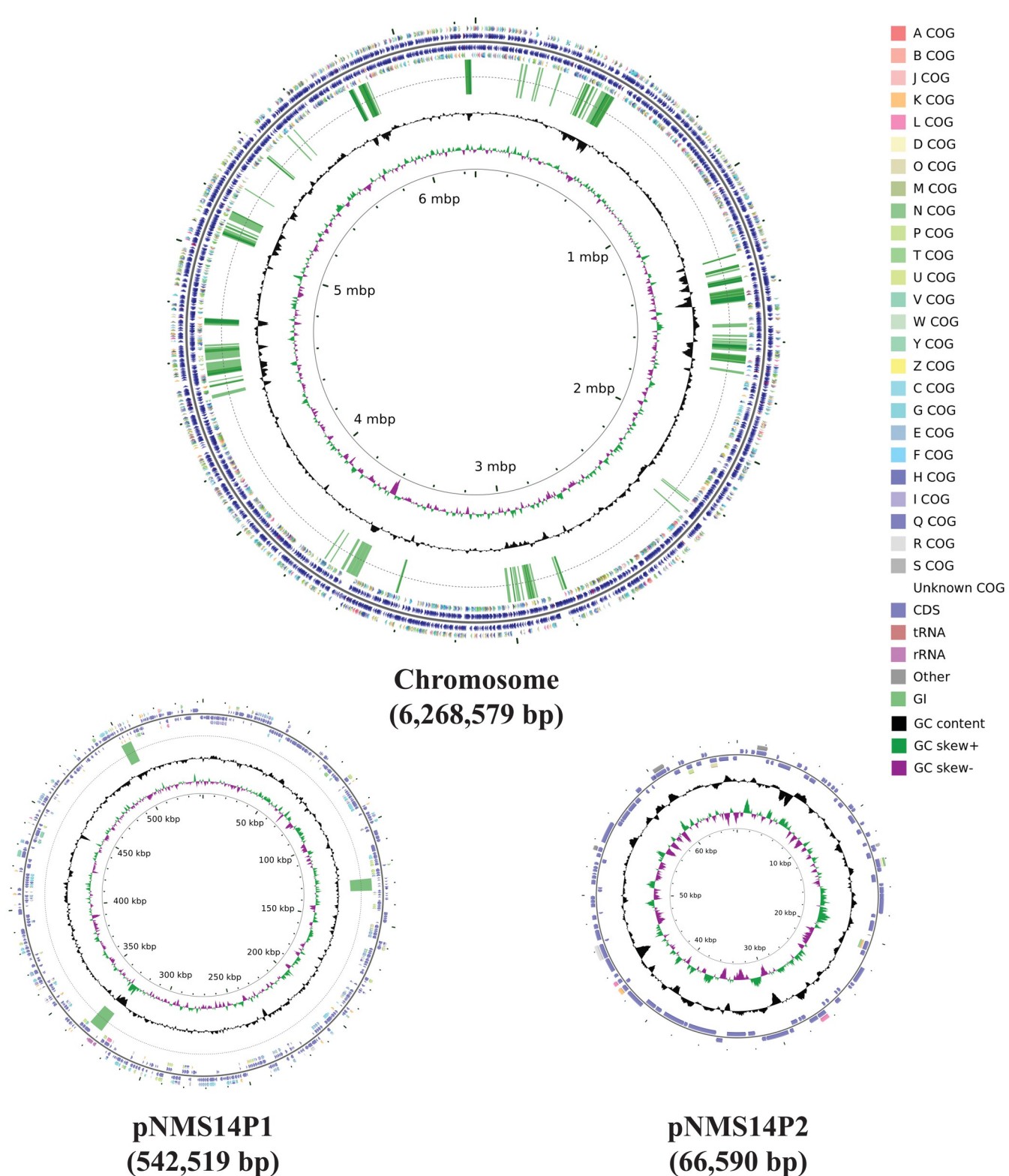

**Fig 2. Schematic representation of the three circular replicons of NMS14P.** The circles from inside to outside: GC skew positive (green) and negative (purple), GC content (black), genomic islands (GIs) (no GI was identified in pNMS14P2), color-coded COG functional categories for reverse strand CDSs, reverse strand sequence features, forward strand sequence features, and color-coded COG functional categories for forward strand CDSs.

**Table 2. Genome structures and general features of NMS14P.**

| | Chromosome | pNMS14P1 | pNMS14P2 |
|---|---|---|---|
| Size (bp) | 6,268,579 | 542,519 | 66,590 |
| G+C content (%) | 71.12 | 70.61 | 64.62 |
| CDS | 5,879 | 496 | 113 |
| CDS assigned to KEGG (BlastKOALA) | 2,649 | 213 | 12 |
| CDS assigned to COG | 3,067 | 240 | 12 |
| CDS assigned to CAZy (dbCAN) | 127 | 4 | - |
| CDS assigned to NCBI-nr | 4,275 | 354 | 28 |
| CDS assigned to RAST server | 3,701 | 322 | 19 |
| Prokka annotation with custom database[a] | 4,840 | 422 | 50 |
| Total annotated CDS | 4,961 | 433 | 54 |
| Unannotated CDS | 918 | 63 | 59 |
| rRNA | 12 | 3 | - |
| tmRNA | 1 | 1 | - |
| tRNA | 73 | - | - |
| misc_RNA | 33 | 1 | 1 |

[a]custom database was constructed by downloading all protein sequences belonging to the genus *Methylobacterium* from the NCBI database.

pNMS14P1, respectively (S6 Table). In addition, antimicrobial resistance (AMR) genes were screened using the CARD pipeline using the strict mode [49]. Evaluation of resistomes in the genome exhibited that NMS1P possessed four copies of *adeF*, an AMR gene in the family of resistance-nodulation-cell division (RND) antibiotic efflux pump (S7 Table).

## Comparative genomic analysis of NMS14P with closely related strains

The genome BLAST comparison of NMS14P against all reference genomes was carried out with CGView Comparison Tools (CCT) [56] using 11 closely related strains with complete, scaffold, or contig-level genomes (Table 1). The BLAST comparison revealed a high level of heterogeneity among the strains, as indicated by light blue colors representing BLAST hit less than 40% similarity (Fig 5). To further analyze the unique regions as well as conserved genes in these genomes, pangenome analysis was performed using the Roary program [55]. The result exhibited a steadily increased number of pangenomes with the sequential addition of each new genome (Fig 6A). In contrast, the size of core genes decreased with each addition of a new genome and finally reached a plateau state with the addition of the fifth genome and stayed nearly constant until the last genome addition (Fig 6A).

A total of 14,291 orthologous groups were discovered, consisting of 3,176 core genes shared by every strain. Several genes involved in methanol oxidation were identified as the core, including the *mxa* gene family (*mxaACDFGIJKL*), *xoxF*, and *pqqBCEL*. In addition, a set of genes (*gckA*, *mdh*, *eno*, *mcl*, *ppc*, *mtkAB*, *hprA*, *AGXT*, and *glyA*) involved in formaldehyde assimilation via serine pathway (KEGG Module: M00346) was also found to be conserved in all *Methylobacterium* genomes in this study. Furthermore, 5,920 dispensable genes and 1,408 genes unique to strain NMS14P were observed, exhibiting the largest number of strain-specific genes among others, while JCM 2831 was the lowest with 11 strain-specific genes (Fig 6B). These NMS14P-unique genes were predominated by unannotated sequences (hypothetical proteins) according to Prokka, KEGG, COG, and custom database annotation (67.19%) (S5 Fig) and might represent candidate putative novel genes, while the rests were genes that are participated in metabolic pathways (56 genes), two-component system (22 genes),

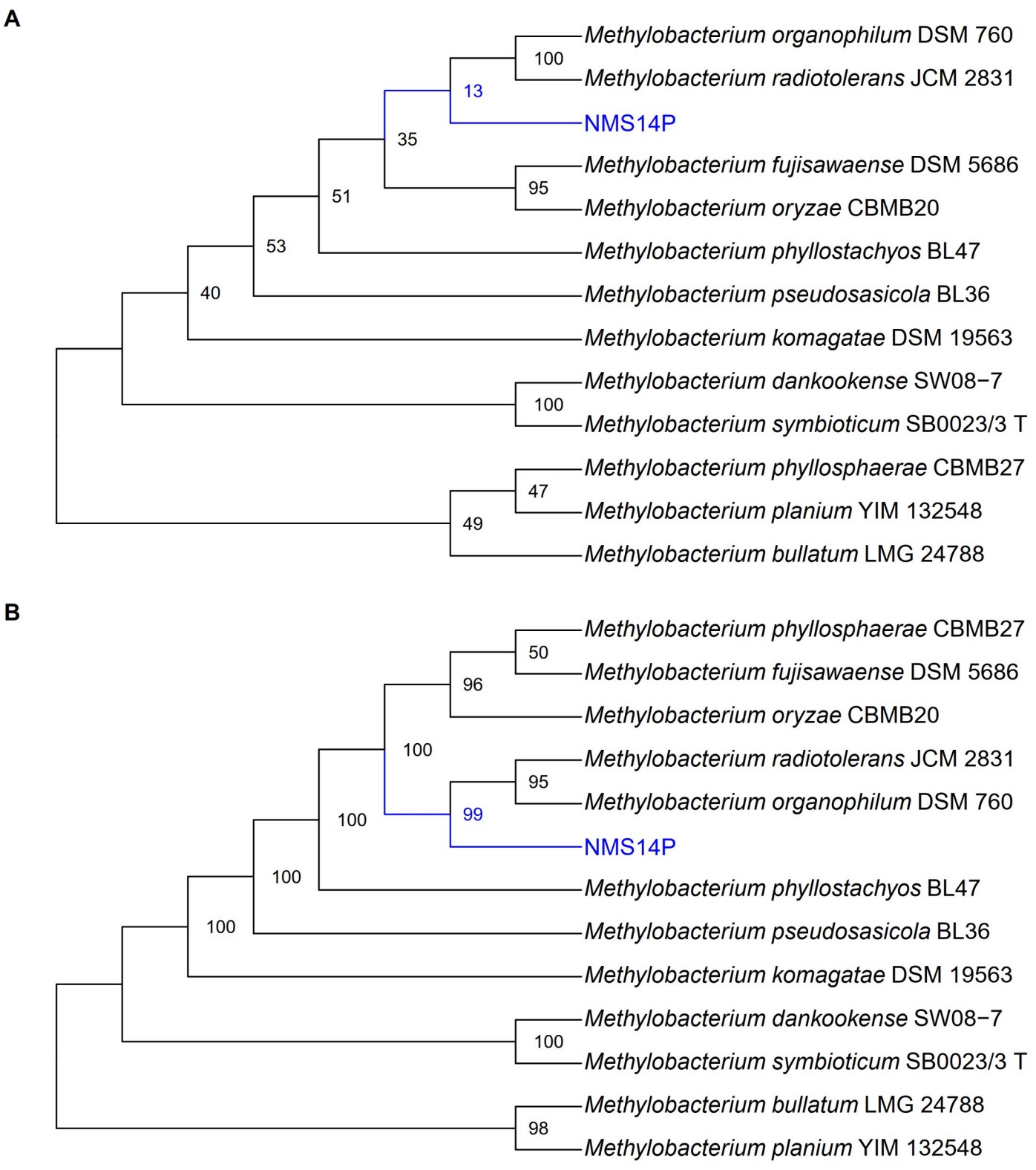

**Fig 3. Phylogenetic analysis.** Phylogenetic trees of NMS14P and other 11 related species were constructed based on (A) 16S rRNA and (B) whole-genome sequences. The maximum likelihood trees were constructed based on 1,000 bootstrap replications.

**Table 3. Pairwise comparisons of NMS14P using dDDH against type strain genomes.**

| Subject strain | dDDH | | | Assembly accession |
|---|---|---|---|---|
| | (d0, in %) | (d4, in %) | (d6, in %) | |
| *Methylobacterium radiotolerans* JCM 2831 | 69.3 | 55.3 | 68.4 | GCA_000019725 |
| *Methylobacterium organophilum* DSM 760 | 70.3 | 55.1 | 69.1 | GCA_003096615 |
| *Methylobacterium oryzae* CBMB20 | 53.9 | 39.7 | 50.8 | GCA_000757795 |
| *Methylobacterium phyllosphaerae* CBMB27 | 55.3 | 39.7 | 51.9 | GCA_900113465 |
| *Methylobacterium fujisawaense* DSM 5686 | 56.0 | 39.4 | 52.3 | - |
| *Methylobacterium phyllostachyos* BL47 | 41.8 | 30.1 | 38.1 | GCA_900103445 |
| *Methylobacterium pseudosasicola* BL36 | 38.5 | 28.3 | 35.1 | GCA_900114535 |
| *Methylobacterium symbioticum* SB0023/3 T | 23.2 | 24.1 | 22.3 | GCA_902141845 |
| *Methylobacterium dankookense* SW08-7 | 24.1 | 24.1 | 23.0 | GCA_902141855 |
| *Methylobacterium komagatae* DSM 19563 | 23.8 | 23.7 | 22.7 | - |
| *Methylobacterium planium* YIM 132548 | 22.1 | 23.4 | 21.3 | GCA_008806345 |
| *Methylobacterium bullatum* LMG 24788 | 17.7 | 22.6 | 17.5 | GCA_014845095 |

microbial metabolism in diverse environments (19 genes), ABC transporter (19 genes), bacterial chemotaxis (15 genes), biosynthesis of secondary metabolites (13 genes), quorum sensing (13 genes), and other pathways with numbers of genes less than 10 (S8 Table).

To annotate the functions of putative novel genes of NMS14P, 946 hypothetical proteins of NMS14P-unique genes were further analyzed using the conserved domain database (CDD) [57]. A total of 56 blast best hits were found, of which 1, 2, and 53 domains originated from the pNMS14P2, pNMS14P1, and chromosome, respectively. Most of the query sequences were assigned to AdoMet_MTases superfamily (3 hits), glycosyltransferase_GTB-type superfamily (2 hits), NepR (2 hits), methyltransferase domain (2 hits), pesticin lyz-like (1 hit), and other domains with only 1 hit (S9 Table).

## Functional analysis of root colonization of NMS14P in multiple hosts

A successful PGPB needs to have competitive capabilities to colonize the target host. In this context, the NMS14P genome contained several genes involved in multiple sugar transporter (*gguA*, *gguB*, and *chvE*; *msmX*, *msmK*, *malK*, *sugC*, *ggtA*, and *msiK*), ribose transporter (*rbsA*, *rbsB*, and *rbsC*), as well as a variety of amino acid transporters such as those for branched-chain amino acid (*livM*, *livK*, *livH*, and *livF*), L-cysteine (*tcyA*, *tcyB*, and *tcyC*), glutamate/aspartate (*gltIJKL/aatJQMP*), and arginine/lysine/histidine (*artR*, *artM*). Moreover, NMS14P also harbored other important genes for the host colonization and rhizocompetence traits, including ABC transporters (115 genes), bacterial chemotaxis (14 genes), flagellar assembly (25 genes), quorum sensing (44 genes), biofilm formation (38 genes), siderophore production (1 gene), and antagonism (1, 1, 3, and 10 genes involved in the gramicidin, pesticin, phenazine, and streptomycin biosynthesis, respectively). Interestingly, some of these genes were exclusively found in NMS14P based on comparative genomic analysis against closely related *Methylobacterium* reference genomes (S6 Fig).

## Identification of plant growth-promoting genes in NMS14P

**Potential capabilities of NMS14P for enhancing nutrient availability to plants.** Nitrogen (N) is an essential microelement for plant growth and development. Plants uptake nitrogen in the form of $NH_4^+$ (ammonium) or $NO_3^-$ (nitrate). NMS14P genome contained a urease gene cluster *ureABCDEFGHJ*, nitrate reductase (*nasAB*), and nitrite reductase (*nirA*),

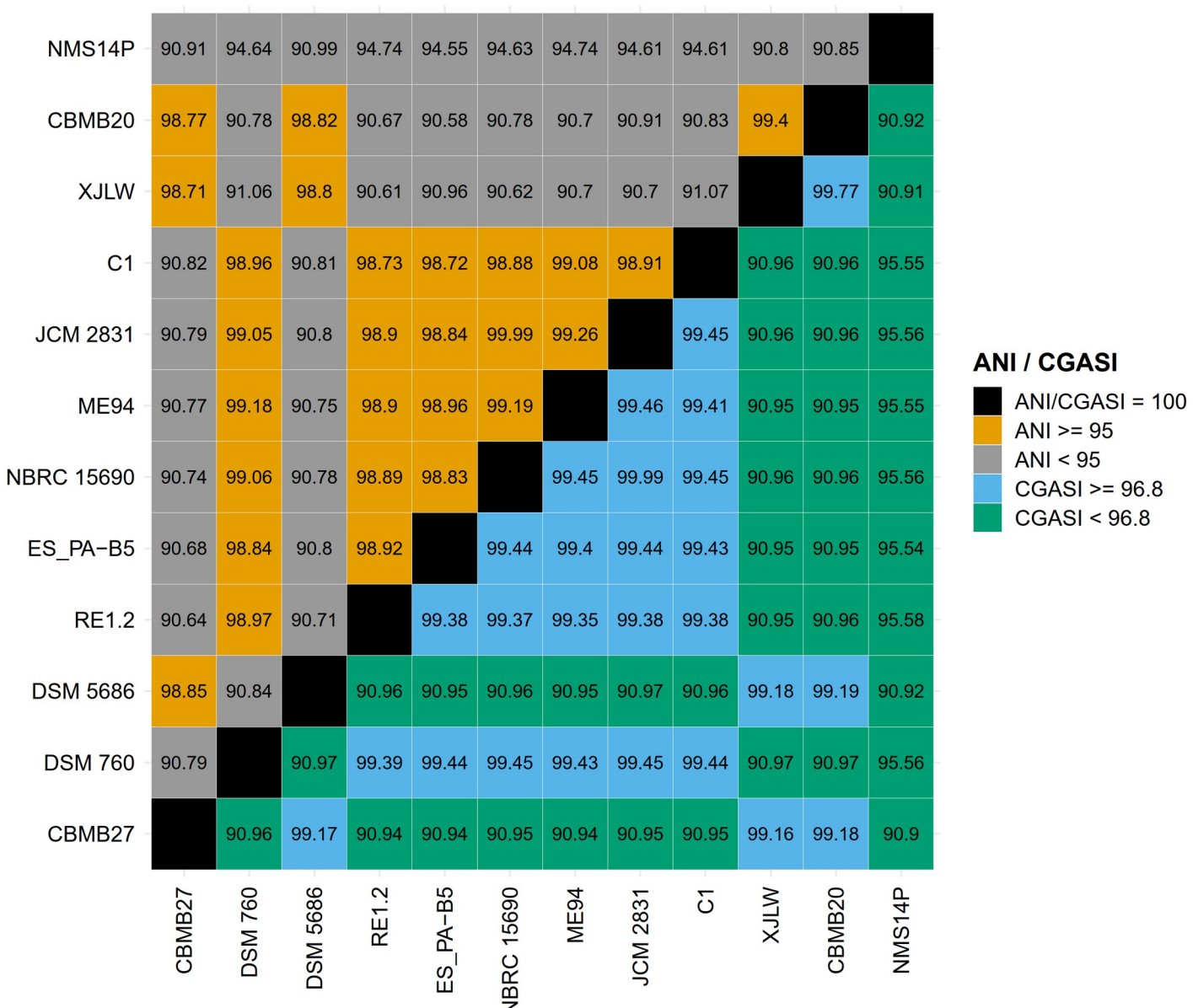

**Fig 4. Comparative genomic analysis.** Pairwise comparisons of NMS14P against closely related species based on average nucleotide identity (ANI, upper triangle) and core genome alignment similarity index (CGASI, lower triangle).

which are involved in the plant-available N support. Nitrate can be converted into ammonia through pathways of assimilatory nitrate reduction (ANRA) and dissimilatory nitrate reduction (DNRA). The ANRA pathway consists of two steps where nitrate is first reduced to nitrite by nitrate reductase (*NasAB*, *NR*, or *NarB*) and is then reduced to ammonia by nitrite reductase (*NirA* or *NIT-6*). The presence of nitrate/nitrite transporter (*nrt*), nitrate reductase (*nasAB*), and ferredoxin-nitrite reductase (*nirA*) genes in the NMS14P genome suggested that this strain could also reduce nitrate into ammonia via the ANRA pathway.

Phosphorus (P) is an indispensable element involved in a plethora of biological and biochemical mechanisms which determine plant growth and yield [58]. Although a large amount

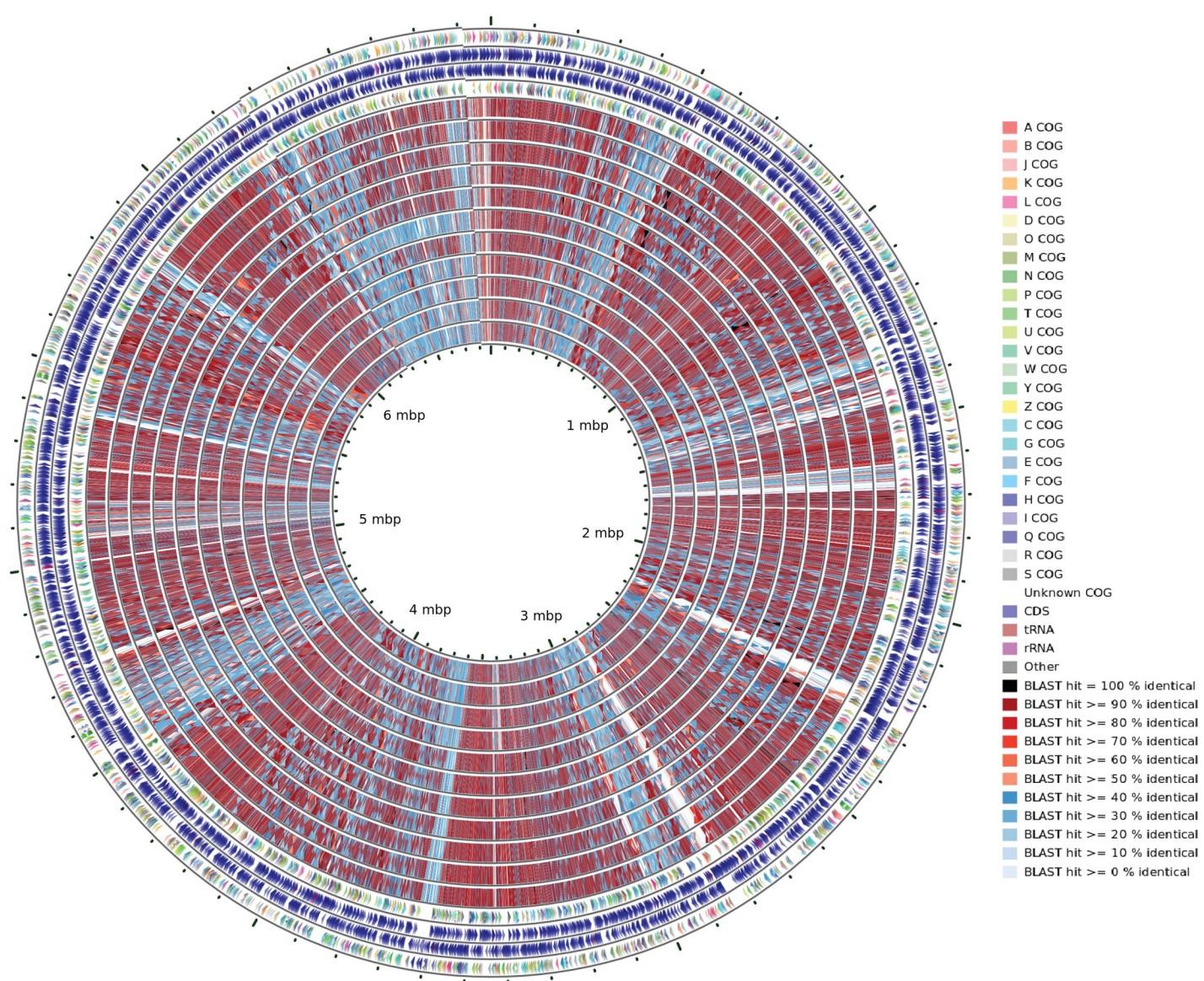

**Fig 5. Whole-genome comparison between NMS14P and other 11 *Methylobacterium* strains.** CGView Comparison Tools (CCT) program was implemented. The outermost track indicates the color-coded COG functional categories for reverse strand CDSs, the next two tracks represent reverse and forward strand sequence features, respectively, and the fourth track indicates color-coded COG functional categories for forward strand CDSs. The next 11 rings show the sequence similarity by BLAST comparisons between the reference genome (*Methylobacterium* sp. NMS14P) and 11 *Methylobacterium* genomes, sequentially as follows: *Methylobacterium radiotolerans* ES_PA-B5, *Methylobacterium radiotolerans* JCM 2831, *Methylobacterium organophilum* DSM 760, *Methylobacterium radiotolerans* ME94, *Methylobacterium radiotolerans* NBRC 15690, *Methylobacterium* sp. C1, *Methylobacterium radiotolerans* RE1.2, *Methylobacterium* sp. XJLW, *Methylobacterium oryzae* CBMB20, *Methylobacterium phyllosphaerae* CBMB27, and *Methylobacterium fujisawaense* DSM 5686.

of P is present in the soil; however, only a small portion is available for plants due to the formation of insoluble organic and inorganic phosphate complexes [59]. Also, the issue of climate change has influenced the availability, acquisition, and translocation of P, creating new challenges in P management [60]. Phosphate (P) solubilization is a crucial attribute to enhance the bioavailability and uptake of essential nutrients for plant growth and development. NMS14P genome contained genes related to P solubilization, especially inorganic P solubilization, including pyrophosphatase (*ppa*), exopolyphosphatase (*ppx*), and glucose dehydrogenase

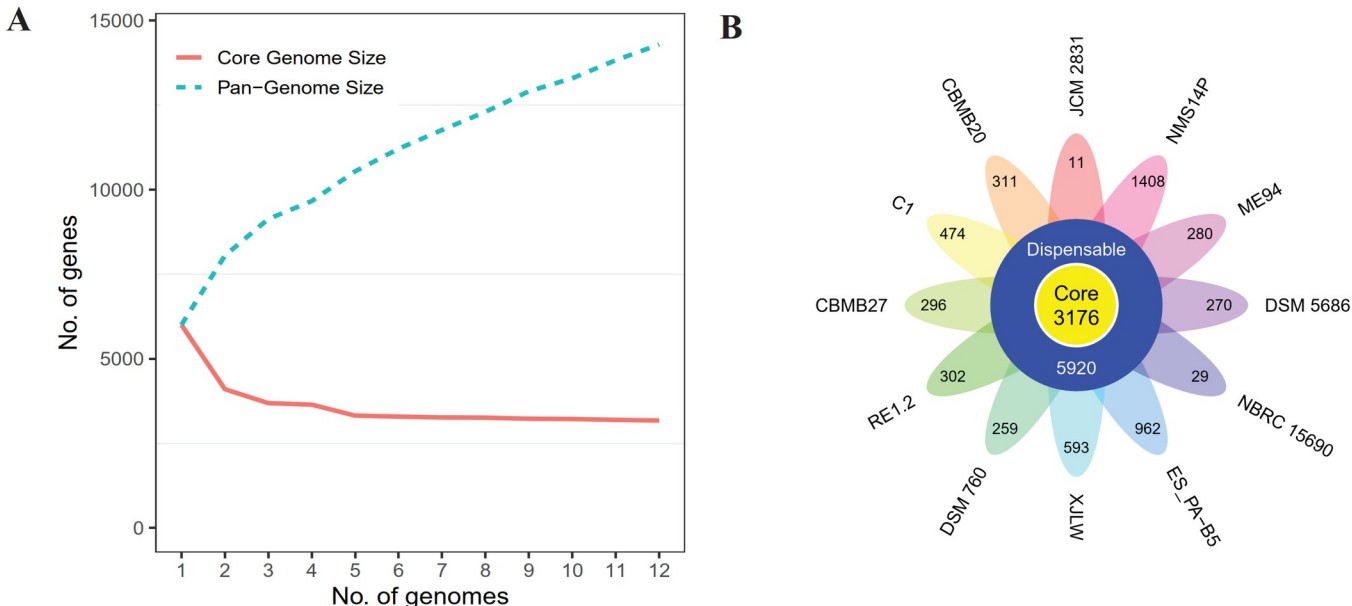

**Fig 6. Core and pangenome analysis of NMS14P.** (A) The number of core and pangenome sizes as a function of the number of genomes added. (B) Flower plot representing the number of core and strain-specific genes of *Methylobacterium* species based on clusters of orthologous groups.

(*gcd*), organic P mineralization, which are the genes in phosphonate and phosphinate metabolism (*phnA*, *phnF*, *phnG*, *phnH*, *phnI*, *phnJ*, *phnK*, *phnL*, *phnM*, *phnN*, *phnP*, *phoD*, *phoN*, and *ugpQ*), P regulation (*phoB*, *phoR*, and *phoU*), and P transportation (*phnC*, *phnD*, *phnE*, *pit*, *pstA*, *pstB*, *pstC*, *pstS*, and *ugpB*). These gene sets indicated that NMS14P could enhance the bioavailability of various recalcitrant P forms for plant uptake in the soil, thus, reducing the dependency on synthetic P fertilizers.

Another vital element for plant growth and development besides N, P, and K is sulfur (S). It plays a role in photosynthesis, respiration, and the formation of chlorophyll and cell membrane structure in plants, which are related to the quality and yield of crop production [61–63]. Plants absorb S as sulfate from the soil, however, soil environments contain sulfate only 1–5% of the total bioavailable sulfur [64]. The sulfur-oxidizing (SOX) system encoded by *SoxACDXYZ* genes was found in the NMS14P genome. The SOX complex is involved in thiosulfate, sulfide, sulfite, and elemental sulfur oxidation and generates sulfate as the final product [65]. The presence of the SOX system supported that NMS14P could increase plant-available S in soil. However, S-sulfosulfanyl-L-cysteine sulfohydrolase (*soxB*) was not found in the genome. The absence of this gene could be further investigated to complete the function of the SOX complex in this NMS14P.

**Potential capabilities of NMS14P in phytohormone production.** Seven genes (*trpABC-DEFG*) involved in the biosynthesis of L-tryptophan, the precursor for auxin biosynthesis, were found in the NMS14P genome. Indole-3-acetaldehyde dehydrogenase, amidase, and nitrilase encoded by *ALDH*, *amiE*, and *nth A-B* genes, respectively, were also identified in the genome, suggesting that NMS14P might be able to produce auxin via indole-3-acetonitrile (IAN) pathway. NMS14P genome also carried the *miaA* gene, which is responsible for the production of zeatin, one type of cytokinins, through isopentenylation of a specific adenine in tRNAs. These phytohormones are not only crucial for various growth and developmental processes in plants but also play a role in the plant-microbe interaction [66].

**Potential capabilities of NMS14P to assist plants withstand environmental stresses.**
Like many other PGPB, NMS14P contained genes that help plants to tolerate environmental
stresses. These were 1-aminocyclopropane-1-carboxylate (ACC) deaminase, trehalose-6-
phosphate synthase (*ostA*), and trehalose-6-phosphate phosphatase (*ostB*). ACC deaminase
can degrade ACC, which is the precursor of ethylene biosynthesis [67], while the last two
genes were involved in trehalose synthesis.

**Potential biocontrol agents of NMS14P for plant disease protection.** Bacterial flagellin
can be recognized by plant flagellin sensing receptors such as FLS2 and FLS3 depending on
plant types, and then triggers plant immune responses and the production of host-defense pro-
teins, which can protect plants from fungal and bacterial infections [68–70]. A total of 25
genes involved in the flagellar assembly, including *FlgBCDEFGHIK*, *FlhAB*, *FliFGIMNPQRY*,
*FlrC*, *MotAB*, and *RpoDN* were detected in the NMS14P genome. In addition, NMS14P also
contained a phenazine biosynthesis gene (*PhzF*).

## Verification of observed plant growth-promoting activities in NMS14P according to the genome information

The potential plant growth-promoting activity of NMS14P was demonstrated by experimental
assays. Pikovskaya's (PKV) agar and the alkaline phosphatase yellow (pNPP) liquid substrate
system were used to test inorganic and organic phosphate solubilizations, respectively. The
clear zone shown only beneath the colony demonstrated that NMS14P could slightly solubilize
tricalcium phosphate (Fig 7A). In the case of organic phosphate, NMS14P could produce alka-
line phosphatase, which could hydrolyze colorless p-nitrophenyl phosphate (pNPP) to p-nitro-
phenol (yellow color) (Fig 7B). Urease activity was measured using a colorimetric method with
phenol red as an indicator. The presence of urease was indicated by the color change of phenol
red from light orange to magenta caused by ammonia production, which increased the pH in
the solution (Fig 7C). 1-aminocyclopropane-1-carboxylic acid (ACC) deaminase activity of
NMS14P was induced by ACC addition in a DF salt minimal medium. Toluenized cell extract
of isolate grown in DF salt minimal medium with ammonium sulfate $(NH_4)_2SO_4$ as the sole
nitrogen source was unable to degrade ACC as shown by the yellow color (Fig 7D). However,
NMS14P cell extracts grown in a DF medium with ACC as the sole nitrogen source could
degrade ACC to ammonia and α-ketobutyrate. After that, α-ketobutyrate derivatized by
2,4-dinitrophenylhydrazine (2,4-DNPH) reagent into α-ketoglutarate 2,4-DNP-hydrazone
and then reacted with sodium hydroxide (NaOH) which changed the color of solution into
brown (Fig 7D). IAA was detected in the NMS14P culture broth (Fig 7E). NMS14P grown in a
half strength of TSB medium with supplementation of 5, 10, and 20 mM of L-tryptophane
showed an increase in IAA production following the rise of tryptophan concentration and
time of incubation. Specifically, the IAA production was 1.58, 2.46, and 3.30 μg/ml at 48 hours
and increased to 5.19, 7.60, and 14.93 μg/ml at 6 days of incubation. These experimental assays
proved that NMS14P had plant growth-promoting activities such as urease, phosphate solubi-
lization, ACC deaminase, and IAA production as were identified from the genome
information.

## Discussion

Plant growth-promoting bacteria (PGPB) promote plant growth through either a direct mech-
anism by facilitating resource acquisition such as N, P, K, S, and other essential minerals or in
an indirect way by lessening the inhibitory effects of numerous pathogens on plant growth and
development in the form of biocontrol agents [67]. Considering their capabilities for

## A. Tricalcium phosphate solubilization

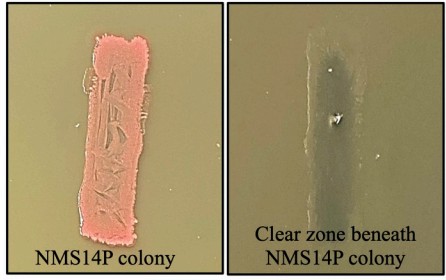

NMS14P colony    Clear zone beneath NMS14P colony

## B. Alkaline phosphatase activity

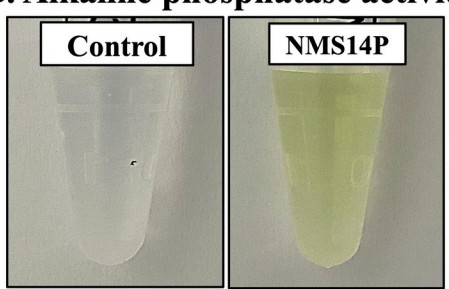

Control    NMS14P

## C. Urease activity

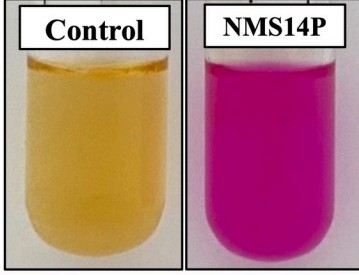

Control    NMS14P

## D. ACC deaminase activity

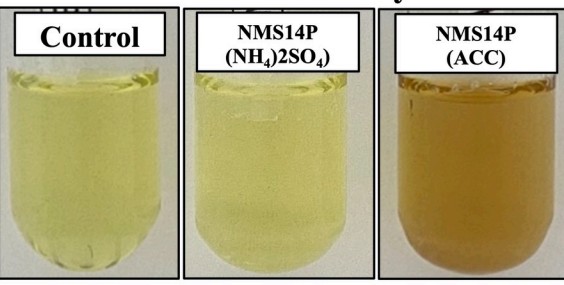

Control    NMS14P (NH₄)2SO₄    NMS14P (ACC)

## E. IAA production

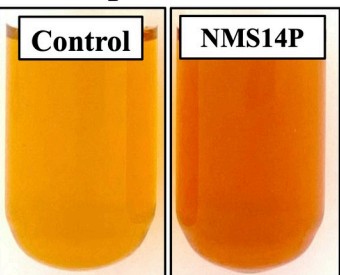

Control    NMS14P

**Fig 7. Experimental assays to confirm plant growth-promoting traits of NMS14P.** The *in vitro* tests include (A) inorganic and (B) organic phosphate solubilizations, (C) urease, (D) ACC deaminase, and (E) IAA production.

improving crop health and productivity in an environment-friendly manner, they serve as an alternative to the application of chemical inputs in agriculture [71, 72].

In the present study, a bacterium (NMS14P) was isolated from the root of a coffee plant (*Coffea arabica*) cultivated organically in Thailand. Taxonomic identification based on 16S rRNA sequence exhibited that this isolate belonged to the genus *Methylobacterium*, and its novelty was further confirmed with genomic and comparative genomic analyses. The results showed that NMS14P had low levels of relatedness with other *Methylobacterium*-type strains in the current database, clarifying that NMS14P is a new methylotrophic bacterial species.

*In vitro* tests using maize, chili, and sugarcane plants inoculated with NMS14P demonstrated that all bioinoculant-treated plants grew and developed better than uninoculated ones, indicating that this isolate had the capability to stimulate plant growth and development. This is supported by the gene contents in the NMS14P genome involving phyto-fertilization and phyto-stimulation, including phosphate solubilization, urea metabolism, sulfur-oxidizing activity, indole-3-acetic acid (IAA) biosynthesis, cytokinins (CKs) production, 1-aminocyclopropane-1-carboxylate (ACC) deaminase activity, and trehalose synthesis. A hypersensitivity reaction (HR) assay was also performed and revealed that this bacterium did not cause any necrosis to the tomato, implying that NMS14P has no potential to infect plants as a phytopathogen. In addition, preliminary safety assessments according to the presence of genes related to virulence factors and antimicrobial resistance (AMR) using various tools and databases indicated that NMS14P is a non-human pathogen; however, before released to the environment, follow-up studies would be required.

NMS14P could promote plant growth and development by enhancing the bioavailability of essential nutrients, including N, P, and S. Although nitrogen-fixing genes (*nifHDK*) were not found in the genome, NMS14P contained a urease gene cluster, in which its activity has been confirmed by *in vitro* test using the colorimetric method. Urease activity plays important roles in protein and nucleotide turnover and urea hydrolysis to improve N availability for soil microbes and plants [73, 74]. It has been reported that the application of PGPB increased soil urease activity, which converts urea fertilizers into ammonia [75], thus promoting plant growth [76]. Also, nitrate reductase and nitrite reductase genes that are involved in the reduction of nitrate into ammonia through the ANRA pathway were found in the NMS14P genome. Increasing rates of the ANRA pathway would have multiple advantages, including reserved N in the soil, reduced N$_2$O emissions, increased farm productivity, reduced water contamination, and reduced use of N fertilizers [77]. Furthermore, the genomic data showed that NMS14P carried genes for P solubilization, thereby contributing to plant phosphate acquisition. The solubilization assay demonstrated that this isolate could solubilize both organic and inorganic phosphate. In addition, the NMS14P genome harbored several genes that participated in the oxidation of thiosulfate, sulfide, sulfite, and elemental sulfur to sulfate as the final product, playing a key role in the cycling of organic sulfate in soils and enhancing plant growth in soils with low S availability [78].

Some PGPB have been known to directly stimulate plant growth through the production of phytohormones, such as indole-3-acetic acid (IAA) [67]. IAA, the main auxin in plants, is a phytohormone that plays an important role in modulating plant growth and development [79]. NMS14P contained genes involved in L-tryptophan biosynthesis and those that encode for indole-3-acetaldehyde dehydrogenase, amidase, and nitrilase, implying that NMS14P might synthesize auxin through indole-3-acetonitrile (IAN) pathway. This result was supported by the previous study that proposed indole-3-acetamide (IAM) and IAN pathways for IAA biosynthesis in *Methylobacterium* sp. 2A [8]. Auxin production from this isolate has been confirmed by *in vitro* test using the Salkowski reagent. Furthermore, the *miaA* gene was also found in the NMS14P genome. It has been reported that *miaA* is required for zeatin production in *Methylobacterium* [80]. Auxin and cytokinin can cause plant cell wall loosening and nutrient leakage from host plants, as well as stimulation of plant growth [81].

Plants are subjected to a broad range of biotic and abiotic stresses that can limit crop growth and productivity. For instance, under stress conditions, the ethylene level in a plant is elevated, leading to the reduction of root and shoot growth [82]. ACC deaminase-producing bacteria can help to decrease ethylene levels, thereby increasing plant growth [83]. NMS14P contained genes encoding for ACC deaminase, in which its activity has been confirmed by the *in vitro* test using a DF salt minimal medium. In addition, NMS14P also harbored genes involved in trehalose biosynthesis. It has been proposed that trehalose produced by beneficial bacteria could trigger the plant-defense system to prevent damage caused by drought [84]. Apart from abiotic stresses, NMS14P possessed genes for synthesizing a globular protein that is a chief constituent of bacterial flagella, called flagellin, which can be recognized by host plants to elicit their immune responses [69]. Likewise, a phenazine biosynthesis gene was also detected in the NMS14P genome. Phenazines are redox-active nitrogen-containing heterocyclic molecules that act as antibiotics and suppress plant pathogenic microbes such as *Rhizoctonia* [85] and *Fusarium* [86]. Altogether, these findings suggested the potential benefit of NMS14P in mitigating the deleterious effects of biotic and abiotic stresses on plant growth and development.

The comparative genomic analysis identified a total of 3,176 core genes, which showed a higher number of core genes in comparison with the previous analysis of nine *Methylobacterium* genomes (2,010 core genes) [87]. As expected, the *mxa* gene family, *xoxF*, and *pqqBCEL* were found in the core genome. *mxa* gene cluster is essential for methanol oxidation using a

pyrroloquinoline quinone (*pqq*)-dependent methanol dehydrogenase (MDH) enzyme, while *xoxF* encodes for the alternative MDH, catalyzing the oxidation of methanol to formaldehyde [88]. The formaldehyde is then either assimilated into cell biomass through the serine cycle or oxidized to $CO_2$ for energy generation in the cofactor tetrahydromethanopterin-dependent pathway [89, 90]. On the other hand, *mxaF* and *xoxF* have been used as biomarkers in environmental studies in that both are highly conserved among methylotrophs [91, 92]. Interestingly, there were two copies of the *act* gene in the NMS14P genome, while only one copy of this gene was found in the other *Methylobaterium* genomes in this study. The *act* gene encodes for the methanol dehydrogenase activator which is involved in the activation of NAD-dependent methanol dehydrogenase (MDH). ACT is an endogenous activator protein, which dramatically enhances the methanol oxidation in *Bacillus methanolicus*, however, the detailed mechanism remains unclear [93]. Two copies of *act* genes in the NMS14P genome might have advantageous attributes in the methanol oxidation activity of this strain, which could be further characterized in the future.

To provide beneficial impacts on the host plants, at first, microbial-based inoculants need to competently colonize the external and/or internal parts of plant tissues, and then establish appropriate interactions with the host [94]. Also, competitive attributes to persist and thrive in a new environment against indigenous microorganisms are highly required [95]. On the other hand, plants secrete sugar and amino acid-enriched metabolites into the soil to attract their microbial counterparts [96]. The composition of root exudates varies across plant species, and specific exudates can act as chemical signals by inhibiting or stimulating the growth of adjacent microbes in the vicinity [97, 98]. Therefore, only microbes that can gain benefits from the utilization of these compounds can colonize and develop mutual interactions with the host plant. Interestingly, NMS14P harbored several genes needed to exploit plant exudates, including those that are involved in multiple sugar transporter, ribose transporter, and numerous amino acid transporters. Moreover, NMS14P also contained other genes necessitated to successfully invade the host plant and outcompete the existing microbial communities, such as ABC transporters, chemotaxis, quorum sensing, biofilm formation, and biosynthesis of secondary metabolites. It has been reported that these pathways play essential roles in the rhizosphere and rhizoplane colonization [94, 99]. Notably, a putative novel gene annotated to encompass the pesticin C-terminal-like domain of uncharacterized proteins (pesticin lyz-like) was found in the chromosome (NMSCH_04528). This subfamily is made up of uncharacterized proteins that have a lysozyme-like domain which is similar to the C-terminal domain of pesticin, an anti-bacterial toxin produced by *Yersinia pestis* used to kill related bacteria of the same niche [100]. Altogether, the presence of numerous transporters related to root exudates, together with host colonization and rhizocompetence traits, might contribute to the broad host range characteristic of NMS14P.

Colonization in some environments might necessitate an increased general genetic flexibility as a prerequisite for successful adaptation and increased fitness to continuously changing growth conditions. Genomic islands (GIs) might offer a selective advantage by enhancing the adaptability and competitiveness of a bacterial species within a habitat under certain growth conditions through large numbers of gene transfer mechanisms [101]. Compared with the previously reported genome, NMS14P contained much more GIs (113 GIs) than *Methylobacterium oryzae* CBMB20, in which 25 GIs were identified in its chromosome [87]. A high number of GIs might be related to the capability of this isolate to colonize and survive in various host plants which possess different environmental conditions. Accordingly, the GI-070 contained genes encoding for linear gramicidin synthase (*lgrBD*), dimodular non-ribosomal peptide synthase (*dhbF*), ferrichrome outer membrane transporter/phage receptor (*fhuA*), and a hypothetical protein. This GI was proposed to be involved in the production of gramicidin

antibiotics. It has been known that linear gramicidin is a pentadecapeptide antibiotic produced via the non-ribosomal pathway [102]. Furthermore, two copies of Na$^+$/H$^+$ antiporter NhaP2 were found in the GI-079. NhaP2 has been reported to play an important role in protecting cells of *Vibrio cholerae* and *Pseudomonas aeruginosa* at low pH [103, 104]. In addition, genes encoding for iron complex transport systems (i.e., ATP-binding protein, permease protein, and substrate-binding protein), ferri-bacillibactin esterase BesA, TonB-dependent siderophore receptor, and regulatory protein PchR were found in the GI-059, implying that this GI was linked to iron acquisition. Iron is an essential element involved in the oxido-reduction mechanisms in the cell. As a result, GIs encoding siderophore systems could be thought of as fitness islands that increase the adaptability of this isolate in the rhizosphere environment [101]. Taken together, these three GIs might also importantly contribute to the colonization ability of NMS14P to various host plants in this study.

## Conclusions

*Methylobacterium* spp. have been known for their functions as a plant growth stimulator in various environmental conditions. The complete genome of a novel species, *Methylobacterium* sp. NMS14P, locally isolated from coffee roots in Thailand, revealed numerous characteristics of PGPB, including urease, phosphate solubilization, phytohormones production, ACC deaminase, and potential biological control agents. Several genes involved in colonization and rhizocompetence have also been reported in this genome, implicating its applicability to a wide array of plant hosts. Based on the *in vivo* test, NMS14P inoculation could promote the growth of both monocot and dicot plants. The results confirmed the potential use of NMS14P as a single-strain *Methylobacterium*-based biofertilizer. However, the application of a synthetic microbial community with known bacterial strains could be more attractive for better PGP activities. As such, the interaction between NMS14P and other bacterial strains as well as target plants could be further investigated for the development of suitable communities, thereby reducing chemical fertilizers for viable sustainable agriculture.

## Supporting information

**S1 Fig. Materials and methods workflow.**
(TIF)

**S2 Fig. Effect of NMS14P treatments on plant growth and biomass.** Comparison of plant biomass of treated- and control (A) maize, (B) chili, and (C) sugarcane at 35-, 75-, and 56-day post-inoculation, respectively.
(TIF)

**S3 Fig. Hypersensitivity reaction (HR) assay.** The hypersensitive response of tomatoes (*Solanum lycopersicum* L.) infiltrated with (A) sterile distilled water, (B) *Pseudomonas aeruginosa*, and (C) NMS14P.
(TIF)

**S4 Fig. Phylogenetic tree constructed based on core genome alignment.**
(TIF)

**S5 Fig. Percentage of hypothetical proteins in the groups of cores, dispensable, and strain-specific genes according to Prokka, KEGG, COG, and custom database annotation.**
(TIF)

**S6 Fig. Strain-specific genes of NMS14P involved in host colonization.**
(TIF)

**S1 Table. Number of reads before and after quality control (QC).**
(XLSX)

**S2 Table. Genome completeness and accuracy.**
(XLSX)

**S3 Table. Genome annotation of NMS14P.**
(XLSX)

**S4 Table. Genomic islands (GIs) identified in the NMS14P genome.**
(XLSX)

**S5 Table. The pathogenic potential of NMS14P evaluated with the PathogenFinder 1.1.**
(XLSX)

**S6 Table. Virulence factors-related genes identified in the NMS14P genome.**
(XLSX)

**S7 Table. Antimicrobial resistance (AMR) genes found in the NMS14P genome.**
(XLSX)

**S8 Table. Strain-specific genes of NMS14P mapped to the KEGG pathway.**
(XLSX)

**S9 Table. Putative novel genes found in the NMS14P genome annotated using a conserved domain database (CDD).**
(XLSX)

**S1 Text. Pairwise alignment of Sanger-derived 16S rRNA sequence with 16S rRNA sequences extracted from the genome assembly.**
(DOCX)

**S1 Graphical abstract.**
(TIF)

## Acknowledgments

We thank the Fungal Biotechnology Laboratory and Systems Biology and Bioinformatics Laboratory, Pilot Plant Development and Training Institute (PDTI), King Mongkut's University of Technology Thonburi (KMUTT) for the facilities. We acknowledge Asst. Prof. Sansanalak Rachdawong (Mitr Phol Group and KMUTT) and Dr. Laddawan Potprommane (PDTI, KMUTT) for their contributions to funding acquisition. We thank Mr. Jirapong Nakhawpech for his help in conducting the experiment part.

## Author Contributions

**Conceptualization:** Jiraporn Jirakkakul, Songsak Wattanachaisaereekul, Supapon Cheevadhanarak, Peerada Prommeenate.

**Data curation:** Jiraporn Jirakkakul, Ahmad Nuruddin Khoiri, Sawannee Sutheeworapong, Supapon Cheevadhanarak.

**Formal analysis:** Jiraporn Jirakkakul, Ahmad Nuruddin Khoiri, Sawannee Sutheeworapong, Kantiya Petsong, Prasobsook Paenkaew, Supapon Cheevadhanarak, Peerada Prommeenate.

**Funding acquisition:** Jiraporn Jirakkakul, Sudarat Dulsawat, Sawannee Sutheeworapong, Songsak Wattanachaisaereekul, Anuwat Tachaleat, Supapon Cheevadhanarak.

**Investigation:** Jiraporn Jirakkakul, Ahmad Nuruddin Khoiri, Thanawat Duangfoo, Sudarat Dulsawat, Sawannee Sutheeworapong, Kantiya Petsong, Songsak Wattanachaisaereekul, Prasobsook Paenkaew, Supapon Cheevadhanarak, Peerada Prommeenate.

**Methodology:** Jiraporn Jirakkakul, Ahmad Nuruddin Khoiri, Thanawat Duangfoo, Sudarat Dulsawat, Sawannee Sutheeworapong, Kantiya Petsong, Songsak Wattanachaisaereekul, Prasobsook Paenkaew, Anuwat Tachaleat, Supapon Cheevadhanarak, Peerada Prommeenate.

**Project administration:** Jiraporn Jirakkakul, Sawannee Sutheeworapong, Supapon Cheevadhanarak, Peerada Prommeenate.

**Resources:** Jiraporn Jirakkakul, Ahmad Nuruddin Khoiri, Thanawat Duangfoo, Sudarat Dulsawat, Sawannee Sutheeworapong, Kantiya Petsong, Songsak Wattanachaisaereekul, Prasobsook Paenkaew, Anuwat Tachaleat, Supapon Cheevadhanarak.

**Software:** Ahmad Nuruddin Khoiri, Sawannee Sutheeworapong, Prasobsook Paenkaew.

**Supervision:** Jiraporn Jirakkakul, Sawannee Sutheeworapong, Songsak Wattanachaisaereekul, Supapon Cheevadhanarak, Peerada Prommeenate.

**Validation:** Jiraporn Jirakkakul, Ahmad Nuruddin Khoiri, Thanawat Duangfoo, Sawannee Sutheeworapong, Kantiya Petsong.

**Visualization:** Jiraporn Jirakkakul, Ahmad Nuruddin Khoiri, Thanawat Duangfoo, Sawannee Sutheeworapong.

**Writing – original draft:** Jiraporn Jirakkakul, Ahmad Nuruddin Khoiri, Thanawat Duangfoo, Sudarat Dulsawat, Supapon Cheevadhanarak, Peerada Prommeenate.

**Writing – review & editing:** Jiraporn Jirakkakul, Ahmad Nuruddin Khoiri, Thanawat Duangfoo, Sudarat Dulsawat, Sawannee Sutheeworapong, Kantiya Petsong, Songsak Wattanachaisaereekul, Prasobsook Paenkaew, Anuwat Tachaleat, Supapon Cheevadhanarak, Peerada Prommeenate.

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
