## [Decision Letter · Decision Letter 0]

13 Jul 2022

PONE-D-22-16071Insights into the genome of a novel Methylobacterium sp. NMS14P, a broad host range plant growth-promoting bacteriumPLOS ONE

Dear Dr. Prommeenate,

Thank you for submitting your manuscript to PLOS ONE. After careful consideration, we feel that it has merit but does not fully meet PLOS ONE’s publication criteria as it currently stands. Therefore, we invite you to submit a revised version of the manuscript that addresses the points raised during the review process.

We look forward to receiving your revised manuscript.

Kind regards,

R. Z. Sayyed, P.D

Academic Editor

PLOS ONE

Journal Requirements:

When submitting your revision, we need you to address these additional requirements. 1. Please ensure that your manuscript meets PLOS ONE's style requirements, including those for file naming. The PLOS ONE style templates can be found at https://journals.plos.org/plosone/s/file?id=wjVg/PLOSOne_formatting_sample_main_body.pdf and https://journals.plos.org/plosone/s/file?id=ba62/PLOSOne_formatting_sample_title_authors_affiliations.pdf
 2. Thank you for stating the following in the Acknowledgments Section of your manuscript:  "AK was financially supported by the “Scholarship for the Development of High Quality Research Graduates in Science and Technology Petchra Pra Jom Klao Ph.D. Research Scholarship (KMUTT – NSTDA) from King Mongkut’s University of Technology Thonburi”. KP was funded by The Program Management Unit for Human Resources & Institutional Development, Research and Innovation [grant number B01F630003]. We thank the Fungal Biotechnology Laboratory and Systems Biology and Bioinformatics Laboratory, Pilot Plant Development and Training Institute (PDTI), King Mongkut’s University of Technology Thonburi (KMUTT) for the facilities. We acknowledge Asst. Prof. Sansanalak Rachdawong (Mitr Phol Group and KMUTT) and Dr. Laddawan Potprommane (PDTI, KMUTT) for their contributions to funding acquisition. We thank Mr. Jirapong Nakhawpech for his help in conducting the experiment part. This research project is supported by Thailand Science Research and Innovation (TSRI) Basic Research Fund: Fiscal year 2022 under project number FRB650048/0164." We note that you have provided funding information that is not currently declared in your Funding Statement. However, funding information should not appear in the Acknowledgments section or other areas of your manuscript. We will only publish funding information present in the Funding Statement section of the online submission form. Please remove any funding-related text from the manuscript and let us know how you would like to update your Funding Statement. Currently, your Funding Statement reads as follows:  "AK was financially supported by the “Scholarship for the Development of High Quality Research Graduates in Science and Technology Petchra Pra Jom Klao Ph.D. Research Scholarship (KMUTT – NSTDA) from King Mongkut’s University of Technology Thonburi”. KP was funded by The Program Management Unit for Human Resources & Institutional Development, Research and Innovation [grant number B01F630003]. This research project is supported by Thailand Science Research and Innovation (TSRI) Basic Research Fund: Fiscal year 2022 under project number FRB650048/0164. The funders had no role in study design, data collection and analysis, decision to publish, or preparation of the manuscript." Please include your amended statements within your cover letter; we will change the online submission form on your behalf.

Reviewers' comments:

Reviewer's Responses to Questions

**Comments to the Author**

1. Is the manuscript technically sound, and do the data support the conclusions?

Reviewer #1: Yes

Reviewer #2: Yes

Reviewer #3: Yes

2. Has the statistical analysis been performed appropriately and rigorously? 

Reviewer #1: Yes

Reviewer #2: Yes

Reviewer #3: N/A

3. Have the authors made all data underlying the findings in their manuscript fully available?

Reviewer #1: No

Reviewer #2: No

Reviewer #3: Yes

4. Is the manuscript presented in an intelligible fashion and written in standard English?

Reviewer #1: No

Reviewer #2: Yes

Reviewer #3: Yes

5. Review Comments to the Author

Reviewer #1: Dear authors,

Thanks for submitting your paper in PlosOne journal. Please note that PlosOne only publishes high-quality papers that provided novel data and scientific materials for academic readers. Therefore, your paper before publishing requires several amendments as mentioned in the following comments. Please carefully answer all comments during revision step, otherwise the journal will not guarantee the handling of further steps.

1. Figure 2: please add more details to circular maps of the genome

2. Figure 3: please reconstruct the phylogenetic tree using the bootstrap method

3. Figure 6: Please compare the pangenome results of NMS14P with other genomes and describes which cluster of genes are similar between all compared genomes

------

4. Please design a flowchart for the M&M section

5. Please add a graphical abstract to the paper

-----

6. Please provide information on laboratory assays. It is essential the respected authors insert some figures of experimental assays within the paper.

7. Please demonstrate which PGP trait was dominant in the studied strain

8. Please add some figures regarding the treatment of maize with the studied bacteria

9. Please supplement the lines of figure 110-112

10. Please supplement the loaded extracted whole genome of the studied bacterium

11. Please provide the value of sequencing depth and add the output of fastq quality control step in supplementary files.

12. Please use PROKKA for genome annotation and add the output of this software in the supplementary files.

13. Please compare the results of genome annotations tools to find the sequence of unique genes that might be ruled out by each software alone.

14. Please compare the genomic sequences of the studied isolate and most similar strains using the BRIG genomics tool.

15. Which novel genes were identified in the studied genome? Please highlight at least 20 novel genes you identified in this genome.

16. In this study, the identification of the studied genome was based on 16S rRNA. Although this method is validated, however, it is better you use the output of genome assembly for subjecting the results to the RAST server. Identification of target species based on genome sequence is accurate by which it can reduce the drawbacks of 16S rRNA characterization.

17. Please identify the major domains of identified novel genes using domain finder tools.

18. Please describe how the inoculant of the studied bacteria can be introduced market. What is the major finding of this study?

19. Please use Orthovenn software to compare the genomic content of annotated genome and similar genomes were used to constructed the phylogenetic trees.

20. Please add the accession number of the submitted genome in the NCBI database. If the respected authors released the genomic data in SAR, please mention the bio-sample accession number.

21. Please supplement the output of 16S rRNA sequencing in supplementary files. Which tool(s) was used for trimming sequenced 16s rRNAs?

22. The discussion sequence is ignored for this paper. Please add an appropriate discussion using updated references to compare your results with previously published papers.

23. Lines 462-472: the authors discussed that they found the genes responsible for ACC deaminase activity of this bacterium. Please using a plate assay to determine whether this bacterium can show such a trait or not? If you use spectroscopic data, please supplement the results.

24. Lines 447-461: please confirm the discussed materials using experimental assays. Although some bacteria have the genes for encoding PGP traits, however, practically they cannot show significant results. Please check this trait using experimental assays and add the results in the supplementary files.

25. The hypersensitivity reaction assay was not determined for this strain. Please show whether it can trigger HR responses or not. It is important to insert the results of this section into the paper. Please use a routine protocol for this assay. Please note that to inject the bacteria into the model plant leaf, try using an injection method not wounding the lower surface of the leaves.

26. Lines 422-434: The respected authors mentioned that the sequenced genome has some genes to solubilize inorganic phosphate. Please confirm this case whether this bacterium can experimentally solubilize inorganic phosphate or not.

27. Genomics islands for possible virulence genes presented in this genome have not been identified. Please use the Genomics Island finder tool to do this section and explain how many genes were identified in the genome annotation step for virulence factors and antibiotic resistance.

28. During the genome assembly step, which software was used for determining k-mers for assembly? Please mention the number of applied k-mers and their size during assembly step.

29. Please use BlastKOALA or GhostKOALA to classify all genomic orthologues you find here.

30. Please use Cytoscape software to perform a gene-ontology analysis for all identified genes discussed in the paper. To do this, you can use the ClueGo module in Cytoscape.

31. Please use different comparative genomics tools to highlight all differences between NMS14P and other genomes. The discussed comparative results in this paper are mainly relying on BLAST. This is not sufficient to highlight the differences between evaluated genomes.

32. Please mention the final size of the genome after the assemblage step.

33. Please separate the results from the discussion section. Please note that the mixture of results and discussion is confusing for academic readers.

34. Please improve some grammatical errors in the paper text. In some sections, it is better the respected authors use shorter sentences to describe their findings

Overall, the manuscript is suitable for publication after amending the above-mentioned comments. Please note that during submitting the revision, prepare an answer sheet to reply all comments line-by line. Please make changes in a different color in the revised version. If the respected authors modify the figures or tables, please mention all changes in the answer sheet.

Reviewer #2: Comments:

This manuscript presents a study on Insights into the genome of a novel Methylobacterium sp. NMS14P, a broad host range plant growth-promoting bacterium. Moreover, the study suffers from several shortfalls including poor presentation. As the comments are numerous to list here, I had marked most of my comments directly on the annotated manuscript.

Comment 1: The first and foremost concern of the study is that the title does not adequately reflect the content of the manuscript. For example, the study about the effects of Methylobacterium sp. NMS14P inoculation on the growth of maize (Zea mays), chili (Capsicum annuum), and sugarcane (Saccharum officinarum L.) is not reflected in the title.

Comment 2: Should be add one paragraph about maize (Zea mays), chili (Capsicum annuum), and sugarcane (Saccharum officinarum L.) in introduction section.

Comment 3: The study has no mechanistic hypothesis.

Comment 4, Line 104: Nutrient broth medium compositions should be written

Comment 5: Much important information is missing in the methods section. The type of soil, soil physicochemical characteristics, variety of maize, chili and sugarcane, plant growth conditions (light, temperature, etc.) are missing.

Comment 6: Discussion should be improved. Why Methylobacterium sp. NMS14P has these positive effects, in which particular part or mechanism is affecting?

Reviewer #3: Brief Summary

The manuscript PONE-D-22-16071 reports the genome characterization of a novel Methylobacterium strain isolated from coffee plant. The study is interesting and could add knowledge to the field. The experiment was conducted with a good design and with valid methodologies. The data handling is suitable, and the quality of the manuscript preparation is appropriate. I would improve some aspects. See specific comments below.

Introduction: The Introduction correctly places the study in the context with a clear statement of the purpose of the research and the working hypotheses being tested. See other comments

Materials and Methods: The authors described with sufficient detail the methods used.

Provide more details on the coffee plant used for the isolation (cultivar and stage of development).

Results and Discussion: Results presentation is clear, and authors correctly discussed the results from the perspective of previous studies. I would expand the results and discussion section by investigating and discussing the genetic asset linked to potassium solubilization activity. See other comments.

Conclusions: The section is appropriate and in line with the findings obtained. The authors should mention some future studies.

Other comments

The references reported could be updated. Several works of 2021 and 2022 could be cited. Follows a list of recent works:

• Applied Sciences, 2021, 11, 3274. https://doi.org/10.3390/app11073274

• Cells, 2021, 10 (6), 1551, https://doi.org/10.3390/cells10061551.

• Frontiers in Microbiology, 2020, 11:580024. https://doi.org/10.3389/fmicb.2020.580024

• Frontiers in Microbiology, 2022, 13:905210, https://doi.org/10.3389/fmicb.2022.905210

• Frontiers in Microbiology, 2022, https://doi.org/10.3389/fmicb.2022.879739

• Microorganisms, 2021, 9, 2511. https://doi.org/10.3390/ microorganisms9122511

• Molecules, 2021, 26, 1894. https://doi.org/10.3390/molecules26071894

• Molecules, 2021, 26, 5758. https://doi.org/10.3390/ molecules26195758

• Molecules, 2021, 26,1569. https://doi.org/10.3390/molecules26061569

• Pathogens, 2021, 10, 1305. https://doi.org/10.3390/pathogens10101305

• Plants 2021, 10, 2436. https://doi.org/10.3390/plants10112436

• Physiology & Mol Biology of Plants, 2020, 26:1847-54

• Scientific Report, 2021:11:22081, https://doi.org/10.1038/s41598-021-01337-9

• Sustainability 2021, 13, 1140. https://doi.org/10.3390/su13031140

• Sustainability, 2021, 13, 5394. https://doi.org/10.3390/su13105394

• Sustainability, 2021, 21,13, 2856. https://doi.org/10.3390/su13052856

6. PLOS authors have the option to publish the peer review history of their article (what does this mean?). If published, this will include your full peer review and any attached files.

Reviewer #1: **Yes: **Hassan Rasouli

Reviewer #2: No

Reviewer #3: No

---

## [Author Response · Author response to Decision Letter 0]

5 Oct 2022

Response to the Editor

ANS:

 We have corrected the manuscript according to the PLOS ONE's style requirements.

"AK was financially supported by the “Scholarship for the Development of High Quality Research Graduates in Science and Technology Petchra Pra Jom Klao Ph.D. Research Scholarship (KMUTT – NSTDA) from King Mongkut’s University of Technology Thonburi”. KP was funded by The Program Management Unit for Human Resources & Institutional Development, Research and Innovation [grant number B01F630003]. We thank the Fungal Biotechnology Laboratory and Systems Biology and Bioinformatics Laboratory, Pilot Plant Development and Training Institute (PDTI), King Mongkut’s University of Technology Thonburi (KMUTT) for the facilities. We acknowledge Asst. Prof. Sansanalak Rachdawong (Mitr Phol Group and KMUTT) and Dr. Laddawan Potprommane (PDTI, KMUTT) for their contributions to funding acquisition. We thank Mr. Jirapong Nakhawpech for his help in conducting the experiment part. This research project is supported by Thailand Science Research and Innovation (TSRI) Basic Research Fund: Fiscal year 2022 under project number FRB650048/0164."

"AK was financially supported by the “Scholarship for the Development of High Quality Research Graduates in Science and Technology Petchra Pra Jom Klao Ph.D. Research Scholarship (KMUTT – NSTDA) from King Mongkut’s University of Technology Thonburi”. KP was funded by The Program Management Unit for Human Resources & Institutional Development, Research and Innovation [grant number B01F630003]. This research project is supported by Thailand Science Research and Innovation (TSRI) Basic Research Fund: Fiscal year 2022 under project number FRB650048/0164. The funders had no role in study design, data collection and analysis, decision to publish, or preparation of the manuscript."

ANS:

Thank you for your suggestions. We have removed our funding statement from the acknowledgment section and put it in the Funding Statement part.

ANS:

Thank you for your suggestions. We have included the amended statements in the new cover letter. 

Response to the Reviewer #1

Thank you for your kind constructive comments and suggestions to improve our manuscript. We have responded to your comments as follows:

1. Figure 2: please add more details to circular maps of the genome

ANS:

We have added more details, i.e., genomic island (GI) regions and the caption as shown in the Fig. 2.

2. Figure 3: please reconstruct the phylogenetic tree using the bootstrap method

ANS:

We have constructed both 16S rRNA and whole genome-based phylogenetic trees using the bootstrap method (1000 bootstraps). The method was described in the caption of the Fig. 3.

3. Figure 6: Please compare the pangenome results of NMS14P with other genomes and describes which cluster of genes are similar between all compared genomes

ANS:

The result of pangenome analysis between NMS14P and other genomes were described in the text [lines 488-490]. We also have described the cluster of genes that are similar between all compared genomes [lines 483-488 and lines 710-717].

4. Please design a flowchart for the M&M section

ANS:

We have made the flowchart for the Materials and Methods section. It is inserted as S1 Figure.

5. Please add a graphical abstract to the paper

ANS:

We have made the graphical abstract and uploaded it as “Graphical_abstract.tif” file. 

6. Please provide information on laboratory assays. It is essential the respected authors insert some figures of experimental assays within the paper.

ANS:

We have added the information on the laboratory assays as provided in the lines 153-214 and as shown in the Fig. 7 and S2 Fig. 

7. Please demonstrate which PGP trait was dominant in the studied strain

ANS:

NMS14P could significantly enhance the growth of maize, chili, and sugarcane. We assumed that it was due to the following PGP traits, including P solubilization, urease, ACC deaminase, and IAA production as demonstrated by in vitro assays and described in the body of the manuscript [lines 663-698]. Moreover, the presence of several genomic islands (GIs) inside the genome of NMS14P, e.g., GI-059, GI-070, and GI-079 which might confer the potential adaptability and competitive traits of this isolate in the rhizosphere environment, could contribute to the dominant trait of this strain. These characteristics involved with its adaptability and competitiveness of the strain; however, further studies need to be done to confirm these activities [lines 750-773]. 

8. Please add some figures regarding the treatment of maize with the studied bacteria

ANS:

We have provided the figure of maize in the Supplementary S1 Fig.

9. Please supplement the lines of figure 110-112

ANS:

Lines 110-112 are the topic of this section (Effects of Methylobacterium sp. NMS14P inoculation on the growth of maize (Zea mays), chili (Capsicum annuum), and sugarcane (Saccharum officinarum L.). The content or description of this section is in the lines 118–151.

10. Please supplement the loaded extracted whole genome of the studied bacterium

ANS:

We have added the values of the loaded extracted whole genome for this studied bacterium, which are 2 �g and 1.2 �g for Illumina and Nanopore sequencing, respectively. [line 220 and line 223]

11. Please provide the value of sequencing depth and add the output of fastq quality control step in supplementary files.

ANS:

We have provided the value of sequencing depth in the manuscript [lines 352-358] and added the outputs of fastq quality control steps in the supplementary S1 Table and please see the details of the quality plots in the following link: https://github.com/NuruddinKhoiry/NMS14P.

12. Please use PROKKA for genome annotation and add the output of this software in the supplementary files.

ANS:

We have used PROKKA for genome annotation. We also have added the PROKKA output in the supplementary S3 Table.

13. Please compare the results of genome annotations tools to find the sequence of unique genes that might be ruled out by each software alone.

ANS:

We have used multiple tools and databases to perform genome annotation, including KEGG, COG, RAST, and NCBI [lines 251-261 and lines 301-304]. The results for all annotations are shown in S3 Table. 

14. Please compare the genomic sequences of the studied isolate and most similar strains using the BRIG genomics tool.

ANS:

Thank you for your suggestion. In this study, we have used the CGView Comparison Tools (CCT) to compare the genomic sequences of the studied isolate with other most similar strains. We have reviewed the BRIG program rigorously and we found that BRIG and CCT are algorithmically similar. Both are BLAST-based comparison. Thus, both CCT and BRIG will provide a similar result. The difference between CCT and BRIG is in terms of the way to draw the output figure. The CCT plot is shown in Fig. 6. With your kind consideration due to time limitation, we would like to keep the CCT plot to be used in Fig. 6.

15. Which novel genes were identified in the studied genome? Please highlight at least 20 novel genes you identified in this genome.

ANS:

We discovered 946 hypothetical proteins which were exclusively found in the NMS14P genome using pangenome analysis. Among these, only 56 CDSs could be functionally annotated with the conserved domain database (CDD) to the group of AdoMet_MTases superfamily, Glycosyltransferase_GTB-type superfamily, NepR, Methyltransferase domain, pesticin lyz-like, and other domains. The details can be found in the manuscript in the lines 498-504. We have also provided the all-candidate novel genes in the Supplementary S9 Table.

16. In this study, the identification of the studied genome was based on 16S rRNA. Although this method is validated, however, it is better you use the output of genome assembly for subjecting the results to the RAST server. Identification of target species based on genome sequence is accurate by which it can reduce the drawbacks of 16S rRNA characterization.

ANS:

We used 16S rRNA gene sequence only for preliminary species identification of the NMS14P isolate. Subsequently, we used genome relative index (OGRI) methods, which uses the whole-genome sequence (output of assembly), as the main approach for bacterial species identification. The OGRI methods includes (i) Type (Strain) Genome Server (TYGS), (ii) core genome alignment method as proposed by Chung et al. (2018), (iii) average nucleotide identity (ANI) algorithm, and (iv) digital DNA-DNA hybridization (dDDH). These methods are highly accurate to conduct species identification as described by Chun et al. (2018) (DOI: 10.1099/ijsem.0.002516). However, we also have added the results from the RAST server annotation per your suggestion [lines 259-260] in the manuscript (Table 2 and Supplementary S3 Table).

17. Please identify the major domains of identified novel genes using domain finder tools.

ANS:

We have identified the major domains of putative novel genes using conserved domain database (CDD) and its search tools. The list of the genes is mentioned in the manuscript in the lines 498-504 and Supplementary S9 Table.

18. Please describe how the inoculant of the studied bacteria can be introduced market. What is the major finding of this study?

ANS: 

Based on the result of this study, NMS14P has the potential to be used as a biofertilizer. Before introducing to the market, however, several processes are required. These are the development of scaling-up procedures for the commercial production of the NMS14P, as well as the formulation for transportation and shelf-life. Moreover, since NMS14P was isolated from the root of a coffee plant in Thailand, which gives an advantage on the geological specificity for Thailand, so it will be suitable for a successful application in the tropical regions.

The major findings of this study are the traits that NMS14P can colonize the host root and establish in the ecosystems, e.g., numerous transporters for utilizing root exudates, host colonization and rhizocompetence genes, and genomic island conferring adaptability traits. In addition, it can promote the overall growth of maize, chili, and sugarcane, which are economical crops of Thailand. 

19. Please use Orthovenn software to compare the genomic content of annotated genome and similar genomes were used to constructed the phylogenetic trees.

ANS:

We have reviewed the features of Orthovenn2, and found its limitation in creating a Venn diagram, as it can accept only 6 genomes and it provides only a few tuning parameters. In contrast, Roary offers more options to be used to adjust and provides more useful information in its outputs. As such, we used the Roary program to compare the genomic content of annotated genome and other similar genomes. 

20. Please add the accession number of the submitted genome in the NCBI database. If the respected authors released the genomic data in SRA, please mention the bio-sample accession number.

ANS:

We have added the BioSample and SRA accession numbers in the manuscript [lines 385-388].

21. Please supplement the output of 16S rRNA sequencing in supplementary files. Which tool(s) was used for trimming sequenced 16s rRNAs?

ANS:

We have provided the sequence of 16S rRNA generated by Sanger sequencing in the S1 Text. 

The tool used to inspect and trim the sequence is Sequence Scanner Software v2.0 (ThermoFisher Scientific) [lines 279-281].

22. The discussion sequence is ignored for this paper. Please add an appropriate discussion using updated references to compare your results with previously published papers.

ANS:

We have separated the Results and Discussion in the manuscript and added some more updated references to compare our results with previously published papers.

23. Lines 462-472: the authors discussed that they found the genes responsible for ACC deaminase activity of this bacterium. Please using a plate assay to determine whether this bacterium can show such a trait or not? If you use spectroscopic data, please supplement the results.

ANS:

We have performed the ACC deaminase activity test using the protocol described by Penrose and Glick (2008). The result was positive and has been included in the manuscript in the lines 618-626 and Fig. 7D.

24. Lines 447-461: please confirm the discussed materials using experimental assays. Although some bacteria have the genes for encoding PGP traits, however, practically they cannot show significant results. Please check this trait using experimental assays and add the results in the supplementary files.

ANS:

We found the genes involved in PGP traits in NMS14P genome, including IAA, zeatin (Cytokinin), and ACC deaminase. The activity tests have been done for IAA and ACC deaminase and the results have been inserted as Fig. 7. However, with laboratory and time limitation, the zeatin activity has not been tested at this time. But it would be interesting to confirm all the PGP traits of NMS14P that involved with plant hormone in the future.

25. The hypersensitivity reaction assay was not determined for this strain. Please show whether it can trigger HR responses or not. It is important to insert the results of this section into the paper. Please use a routine protocol for this assay. Please note that to inject the bacteria into the model plant leaf, try using an injection method not wounding the lower surface of the leaves.

ANS:

Thank you for your suggestion about the hypersensitivity reaction assay. However, we cannot perform the assay at this time as we do not have the model plants ready in our laboratory. We would be able to continue this work in the future for hypersensitivity reaction test of NMS14P when the model plants are available. 

26. Lines 422-434: The respected authors mentioned that the sequenced genome has some genes to solubilize inorganic phosphate. Please confirm this case whether this bacterium can experimentally solubilize inorganic phosphate or not.

ANS:

We have performed the inorganic phosphate solubilization test using Pikovskaya’s (PVK) agar plate according to Pikovskaya (1948). The result was positive and has been included in the manuscript in Fig. 7A and lines 610-613.

27. Genomics islands for possible virulence genes presented in this genome have not been identified. Please use the Genomics Island finder tool to do this section and explain how many genes were identified in the genome annotation step for virulence factors and antibiotic resistance.

ANS:

We have performed genomic islands (GI) prediction with IslandViewer 4. The results were presented in S4 Table. Likewise, the pathogenic potential, virulence factors, and antimicrobial resistance (AMR) genes were detected using PathogenFinder, VFDB, and CARD, respectively. [lines 447-469]

28. During the genome assembly step, which software was used for determining k-mers for assembly? Please mention the number of applied k-mers and their size during assembly step.

ANS:

In this manuscript, Trycycler pipeline (https://github.com/rrwick/Trycycler) was used to perform the genome assembly. More specifically, we used Flye, Miniasm, Minipolish, and Raven assemblers with their default settings to generate initial genome assemblies. Thus, we did not use any specific program to determine k-mers for assembly. 

29. Please use BlastKOALA or GhostKOALA to classify all genomic orthologues you find here.

ANS:

We have re-annotated all genomic orthologues with GhostKOALA. We also performed annotation with DIAMOND BLASTx against COG, and custom Methylobacterium databases downloaded from RefSeq NCBI. [lines 301-304]

30. Please use Cytoscape software to perform a gene-ontology analysis for all identified genes discussed in the paper. To do this, you can use the ClueGo module in Cytoscape.

ANS:

We can’t do this analysis because ClueGO requires a reference data “Marker List” to perform the analysis. However, NMS14P is a new species, and the reference data of its closely related taxa are not in the download-able list of ClueGO database yet. We have contacted the creator of the program (Dr. Gabriela Bindea) to help us by email. Unfortunately, until this response letter is written (29 September 2022), we haven’t gotten any reply from her yet. 

31. Please use different comparative genomics tools to highlight all differences between NMS14P and other genomes. The discussed comparative results in this paper are mainly relying on BLAST. This is not sufficient to highlight the differences between evaluated genomes.

ANS:

Thank you for your suggestion. In this study, the differences between NMS14P and other genomes have been analyzed using many tools and methods, including average nucleotide identity (ANI) with FastANI, DNA-DNA hybridization (DDH) with GGDC, phylogenomic with TYGS, core genome similarity index (CGSI) proposed by Chung et al. (2018), CGView comparison tools (CCT), and pangenomic with Roary. 

32. Please mention the final size of the genome after the assemblage step.

ANS:

We have mentioned the final size of the genome after the assemblage step in the Table 2. In brief, the size for chromosome is 6,268,579 bp, pNMS14P1 is 542,519 bp, and pNMS14P2 is 66,590 bp.

33. Please separate the results from the discussion section. Please note that the mixture of results and discussion is confusing for academic readers.

ANS:

We have separated the Results and Discussion parts in the manuscript.

34. Please improve some grammatical errors in the paper text. In some sections, it is better the respected authors use shorter sentences to describe their findings

ANS:

 We have corrected the grammatical errors in the manuscript.

Overall, the manuscript is suitable for publication after amending the above-mentioned comments. Please note that during submitting the revision, prepare an answer sheet to reply all comments line-by-line. Please make changes in a different color in the revised version. If the respected authors modify the figures or tables, please mention all changes in the answer sheet.

ANS:

Thank you for your comments. The manuscript has been revised and all the changes have been labeled with red color and indicated by line number(s) in this letter. 

Response to Reviewer #2

This manuscript presents a study on Insights into the genome of a novel Methylobacterium sp. NMS14P, a broad host range plant growth-promoting bacterium. Moreover, the study suffers from several shortfalls including poor presentation. As the comments are numerous to list here, I had marked most of my comments directly on the annotated manuscript.

ANS:

Thank you for your comments; however, the annotated manuscript with your comments has not been found during the time of revision. Please consider the revision version with this submission. 

1. Comment 1: The first and foremost concern of the study is that the title does not adequately reflect the content of the manuscript. For example, the study about the effects of Methylobacterium sp. NMS14P inoculation on the growth of maize (Zea mays), chili (Capsicum annuum), and sugarcane (Saccharum officinarum L.) is not reflected in the title.

ANS:

We have changed the title of the manuscript to “Insights into the genome of Methylobacterium sp. NMS14P, a novel bacterium for growth promotion of maize, chili, and sugarcane”

2. Comment 2: Should be add one paragraph about maize (Zea mays), chili (Capsicum annuum), and sugarcane (Saccharum officinarum L.) in introduction section.

ANS:

We have added the paragraph about maize (Zea mays), chili (Capsicum annuum), and sugarcane (Saccharum officinarum L.) in introduction section [Lines 81-84].

3. Comment 3: The study has no mechanistic hypothesis.

ANS:

In this study, the newly identified Methylobacterium sp. NMS14P has been isolated and studied according to its PGP traits. We have added the paragraph in the “Introduction” [lines 77-89] for the hypothesis, as follows. In this study, Methylobacterium spp. were isolated from organic coffee roots by imprinting methods. One of the isolates designated as Methylobacterium sp. NMS14P (hereafter referred to as NMS14P) revealed significant PGP activities as observed from the various plant phenotypes (maize, chili, and sugarcane), which included root development, height, and total biomass. Maize, chili, and sugarcane are economical crops of Thailand, which require a lot of chemical fertilizers and pesticides to increase their productivity. Therefore, the success in the application of NMS14P as a potential native PGPB could provide an alternative way for sustainable agriculture management. To evaluate the functional capabilities of NMS14P as a PGPB, the whole genome of NMS14P was sequenced and comparatively analyzed with 11 closely related Methylobacterium genomes. The overall genomic information of NMS14P supports and provides insight into the molecular mechanisms underlying the plant growth-promoting characteristics of NMS14P, which could be applied to the development of either a single or consortium of Methylobacterium-based biofertilizers in the future.

4. Comment 4, Line 104: Nutrient broth medium compositions should be written.

ANS:

We used the commercial nutrient broth (NB) medium from Difco, USA.

5. Comment 5: Much important information is missing in the methods section. The type of soil, soil physicochemical characteristics, variety of maize, chili and sugarcane, plant growth conditions (light, temperature, etc.) are missing.

ANS:

Thank you for your suggestion. The plants were grown on peat moss and perlite at a 1:1 (v/v) ratio. We have added the missing information in the Materials and Methods section [lines 118-142]. 

6. Comment 6: Discussion should be improved. Why Methylobacterium sp. NMS14P has these positive effects, in which particular part or mechanism is affecting?

ANS:

We have improved the discussion part and explained that NMS14 could promote plant growth by enhancing the bioavailability of essential nutrients, including N, P, and S as discussed in the lines 662-680 and by producing IAA [lines 681-689] and ACC deaminase [693-698].

Response to Reviewer #3

The manuscript PONE-D-22-16071 reports the genome characterization of a novel Methylobacterium strain isolated from coffee plant. The study is interesting and could add knowledge to the field. The experiment was conducted with a good design and with valid methodologies. The data handling is suitable, and the quality of the manuscript preparation is appropriate. I would improve some aspects. See specific comments below.

ANS:

Thank you for your kind constructive comments and suggestions to improve our manuscript. We have responded to your comments as follows:

1. Introduction: The Introduction correctly places the study in the context with a clear statement of the purpose of the research and the working hypotheses being tested. See other comments

ANS: 

Thank you for your comments. 

2. Materials and Methods: The authors described with sufficient detail the methods used. Provide more details on the coffee plant used for the isolation (cultivar and stage of development).

ANS:

The coffee plant used for the isolation of NMS14P was Coffea arabica, Chiang Mai 80 variety, in the vegetative growth phase. The details are in the lines 93-94.

3. Results and Discussion: Results presentation is clear, and authors correctly discussed the results from the perspective of previous studies. I would expand the results and discussion section by investigating and discussing the genetic asset linked to potassium solubilization activity. See other comments.

ANS:

We have tested the capability of NMS14P isolate to solubilize potassium (K) using in vitro test with Aleksandrov medium, however, the result was negative. Also, we could not find genes related to K solubilization in its genome.

4. Conclusions: The section is appropriate and in line with the findings obtained. The authors should mention some future studies.

ANS:

We have mentioned some future studies in the manuscript as followed “Based on the in vivo test, NMS14P inoculation could promote the growth of both monocot and dicot plants. The results confirmed the potential use of NMS14P as a single strain Methylobacterium-based biofertilizer. However, the application of a synthetic microbial community, with known bacterial strains could be more attractive for better PGP activities. As such, the interaction between NMS14P and other bacterial strains, as well as target plants, could be further investigated for the development of suitable communities, thereby reducing chemical fertilizers for viable sustainable agriculture. [lines 782-788].

5. Other comments

The references reported could be updated. Several works of 2021 and 2022 could be cited. Follows a list of recent works:

• Applied Sciences, 2021, 11, 3274. https://doi.org/10.3390/app11073274

• Cells, 2021, 10 (6), 1551, https://doi.org/10.3390/cells10061551

• Frontiers in Microbiology, 2020, 11:580024. https://doi.org/10.3389/fmicb.2020.580024

• Frontiers in Microbiology, 2022, 13:905210, https://doi.org/10.3389/fmicb.2022.905210

• Frontiers in Microbiology, 2022, https://doi.org/10.3389/fmicb.2022.879739

• Microorganisms, 2021, 9, 2511. https://doi.org/10.3390/microorganisms9122511

• Molecules, 2021, 26, 1894. https://doi.org/10.3390/molecules26071894

• Molecules, 2021, 26, 5758. https://doi.org/10.3390/molecules26195758

• Molecules, 2021, 26,1569. https://doi.org/10.3390/molecules26061569

• Pathogens, 2021, 10, 1305. https://doi.org/10.3390/pathogens10101305

• Plants 2021, 10, 2436. https://doi.org/10.3390/plants10112436

• Physiology & Mol Biology of Plants, 2020, 26:1847-54

• Scientific Report, 2021:11:22081, https://doi.org/10.1038/s41598-021-01337-9

• Sustainability 2021, 13, 1140. https://doi.org/10.3390/su13031140

• Sustainability, 2021, 13, 5394. https://doi.org/10.3390/su13105394

• Sustainability, 2021, 21,13, 2856. https://doi.org/10.3390/su13052856

ANS:

We have used some references that you suggested to update our references. These are reference numbers 70, 71, and 82.

---

## [Decision Letter · Decision Letter 1]

2 Dec 2022

PONE-D-22-16071R1Insights into the genome of Methylobacterium sp. NMS14P, a novel bacterium for growth promotion of maize, chili, and sugarcanePLOS ONE

Dear Dr. Prommeenate,

Thank you for submitting your manuscript to PLOS ONE. After careful consideration, we feel that it has merit but does not fully meet PLOS ONE’s publication criteria as it currently stands. Therefore, we invite you to submit a revised version of the manuscript that addresses the points raised during the review process, particularly comments from Reviewer 1.

We look forward to receiving your revised manuscript.

Kind regards,

Ying Ma, Ph.D.

Academic Editor

PLOS ONE

Journal Requirements:

Reviewers' comments:

Reviewer's Responses to Questions

**Comments to the Author**

1. If the authors have adequately addressed your comments raised in a previous round of review and you feel that this manuscript is now acceptable for publication, you may indicate that here to bypass the “Comments to the Author” section, enter your conflict of interest statement in the “Confidential to Editor” section, and submit your "Accept" recommendation.

Reviewer #1: All comments have been addressed

Reviewer #2: All comments have been addressed

Reviewer #3: All comments have been addressed

2. Is the manuscript technically sound, and do the data support the conclusions?

Reviewer #1: Partly

Reviewer #2: Yes

Reviewer #3: Yes

3. Has the statistical analysis been performed appropriately and rigorously? 

Reviewer #1: Yes

Reviewer #2: Yes

Reviewer #3: Yes

4. Have the authors made all data underlying the findings in their manuscript fully available?

Reviewer #1: Yes

Reviewer #2: Yes

Reviewer #3: (No Response)

5. Is the manuscript presented in an intelligible fashion and written in standard English?

Reviewer #1: Yes

Reviewer #2: Yes

Reviewer #3: Yes

6. Review Comments to the Author

Reviewer #1: Dear authors

Thanks for your answers. However, one of the most important features of this work still requires further validation. As I previously addressed in my comments, the authors should provide HR assay output for the studied strains. Because these strains have the potential to introduce into agricultural markets by the means of biofertilizers, therefore, the respected authors should test whether the studied bacteria could cause hypersensitivity reactions in plant models or not. Please use one of the below plants to conduct this assay. Without this validation assay, the current results cannot be published.

1- Tomato

2- Maize

3- Tobacco

4- Geraniums

5- Sugarcane

While the respected authors conduct this assay, I highly recommend you to prepare high-resolution figures of HR assays to provide detailed information for the academic readers of this journal.

Reviewer #2: The authors have sufficiently improved the manuscript. The manuscript should be accepted for publication.

Reviewer #3: The authors correctly addressed all my previous comments. The manuscript is signficantly improved. I have no further suggestions.

7. PLOS authors have the option to publish the peer review history of their article (what does this mean?). If published, this will include your full peer review and any attached files.

Reviewer #1: **Yes: **Rasouli. H

Reviewer #2: No

Reviewer #3: No

---

## [Author Response · Author response to Decision Letter 1]

23 Jan 2023

Response to the Reviewer #1

Reviewer #1: Dear authors

Thanks for your answers. However, one of the most important features of this work still requires further validation. As I previously addressed in my comments, the authors should provide HR assay output for the studied strains. Because these strains have the potential to introduce into agricultural markets by the means of biofertilizers, therefore, the respected authors should test whether the studied bacteria could cause hypersensitivity reactions in plant models or not. Please use one of the below plants to conduct this assay. Without this validation assay, the current results cannot be published.

1- Tomato

2- Maize

3- Tobacco

4- Geraniums

5- Sugarcane

While the respected authors conduct this assay, I highly recommend you to prepare high-resolution figures of HR assays to provide detailed information for the academic readers of this journal.

ANS:

Thank you for your constructive suggestions.

Per your suggestions, we have performed the HR assay using tomatoes (Solanum lycopersicum L.) as a plant model. The method used for the HR test was injection underneath the leaf while supporting the leaf with a finger as explained by Huang et al. (1988) (lines 216-224). The HR assay result for NMS14P was negative, while Pseudomonas aeruginosa (positive control) caused necrotic responses on the injected leaves. The result has been incorporated into the manuscript in lines 342-344 (Results) and 647-650 (Discussion). The high-resolution figures of HR assays were provided as S3 Fig. 

Response to the Reviewer #2

Reviewer #2: The authors have sufficiently improved the manuscript. The manuscript should be accepted for publication.

ANS:

Thank you for your comments and suggestions to improve this manuscript.

Response to the Reviewer #3

Reviewer #3: The authors correctly addressed all my previous comments. The manuscript is significantly improved. I have no further suggestions.

ANS:

Thank you for your comments and suggestions to improve this manuscript.

---

## [Decision Letter · Decision Letter 2]

25 Jan 2023

Insights into the genome of Methylobacterium sp. NMS14P, a novel bacterium for growth promotion of maize, chili, and sugarcane

PONE-D-22-16071R2

Dear Dr. Prommeenate,

We’re pleased to inform you that your manuscript has been judged scientifically suitable for publication and will be formally accepted for publication once it meets all outstanding technical requirements.

Kind regards,

Ying Ma, Ph.D.

Academic Editor

PLOS ONE

Additional Editor Comments (optional):

Reviewers' comments:

Reviewer's Responses to Questions

**Comments to the Author**

1. If the authors have adequately addressed your comments raised in a previous round of review and you feel that this manuscript is now acceptable for publication, you may indicate that here to bypass the “Comments to the Author” section, enter your conflict of interest statement in the “Confidential to Editor” section, and submit your "Accept" recommendation.

Reviewer #1: All comments have been addressed

2. Is the manuscript technically sound, and do the data support the conclusions?

Reviewer #1: Yes

3. Has the statistical analysis been performed appropriately and rigorously? 

Reviewer #1: Yes

4. Have the authors made all data underlying the findings in their manuscript fully available?

Reviewer #1: Yes

5. Is the manuscript presented in an intelligible fashion and written in standard English?

Reviewer #1: Yes

6. Review Comments to the Author

Reviewer #1: Dear authors,

Great congratulations on the new version of your manuscript. I have checked the manuscript content and the HR assay was accurately conducted in this study. I have no further comments and the manuscript is suitable for publication.

7. PLOS authors have the option to publish the peer review history of their article (what does this mean?). If published, this will include your full peer review and any attached files.

Reviewer #1: **Yes: **Rasouli. H

---

## [Editor Report · Acceptance letter]

27 Jan 2023

PONE-D-22-16071R2 

Insights into the genome of Methylobacterium sp. NMS14P, a novel bacterium for growth promotion of maize, chili, and sugarcane 

Dear Dr. Prommeenate:

I'm pleased to inform you that your manuscript has been deemed suitable for publication in PLOS ONE. Congratulations! Your manuscript is now with our production department. 

Kind regards, 

on behalf of

Dr. Ying Ma 

Academic Editor

PLOS ONE